# The neurotransmitter calcitonin gene-related peptide shapes an immunosuppressive microenvironment in medullary thyroid cancer

Yingtong Hou[1,7], Bo Lin[2,7], Tianyi Xu[1,7], Juan Jiang[3,7], Shuli Luo[1,7], Wanna Chen[2,7], Xinwen Chen[1], Yuanqi Wang[4], Guanrui Liao[4], Jianping Wang[5], Jiayuan Zhang[1], Xuyang Li[1], Xiao Xiang[4], Yubin Xie[3], Ji Wang[3], Sui Peng[3,6], Weiming Lv[2], Yihao Liu[6,8] ✉ & Haipeng Xiao[1,8] ✉

Neurotransmitters are key modulators in neuro-immune circuits and have been linked to tumor progression. Medullary thyroid cancer (MTC), an aggressive neuroendocrine tumor, expresses neurotransmitter calcitonin gene-related peptide (CGRP), is insensitive to chemo- and radiotherapies, and the effectiveness of immunotherapies remains unknown. Thus, a comprehensive analysis of the tumor microenvironment would facilitate effective therapies and provide evidence on CGRP's function outside the nervous system. Here, we compare the single-cell landscape of MTC and papillary thyroid cancer (PTC) and find that expression of CGRP in MTC is associated with dendritic cell (DC) abnormal development characterized by activation of cAMP related pathways and high levels of Kruppel Like Factor 2 (KLF2), correlated with an impaired activity of tumor infiltrating T cells. A CGRP receptor antagonist could offset CGRP detrimental impact on DC development in vitro. Our study provides insights of the MTC immunosuppressive microenvironment, and proposes CGRP receptor as a potential therapeutic target.

Neuroimmunology is an emerging field that links two principal systems, the nervous system and immunity[1]. Recent advances in neuroimmunology revealed a tumor-promoting function of neurotransmitters in cancer progression. For example, nociceptor neurons in melanoma promoted the exhaustion of tumor-infiltrating T cells by releasing calcitonin gene-related peptide (CGRP)[2]. In oral mucosa carcinoma, nociceptive nerves can induce cytoprotective autophagy of tumor cells through CGRP production[3]. Although the

role of neurotransmitters in modulating tumor progression is gaining increasing attention, key evidence is mainly from in vitro and animal studies at the moment, probably due to difficulties in obtaining human samples and the lack of an adequate disease model.

Neuroendocrine tumors (NETs) could be such a suitable model to study the potential role of neurotransmitters in human tumors. NETs are epithelial-derived malignancies commonly found in the lung, gastrointestinal tract and thyroid, that can secrete various types of

[1]Department of Endocrinology, The First Affiliated Hospital, Sun Yat-Sen University, Guangzhou, China. [2]Department of Thyroid Surgery, The First Affiliated Hospital, Sun Yat-sen University, Guangzhou, China. [3]Institute of Precision Medicine, The First Affiliated Hospital, Sun Yat-sen University, Guangzhou, China. [4]Department of Liver Surgery, Center of Hepato-Pancreato-Biliary Surgery, The First Affiliated Hospital, Sun Yat-sen University, Guangzhou, China. [5]Department of Gastroenterology and Hepatology, The First Affiliated Hospital, Sun Yat-sen University, Guangzhou, China. [6]Clinical Trials Unit, The First Affiliated Hospital, Sun Yat-sen University, Guangzhou, China. [7]These authors contributed equally: Yingtong Hou, Bo Lin, Tianyi Xu, Juan Jiang, Shuli Luo, Wanna Chen. [8]These authors jointly supervised this work: Yihao Liu, Haipeng Xiao. ✉e-mail: liuyih3@mail2.sysu.edu.cn; xiaohp@mail.sysu.edu.cn

hormones and neurotransmitters, such as insulin, 5-hydroxytryptamine, calcitonin, and CGRP[4]. Medullary thyroid cancer (MTC) is a type of NETs that originates from thyroid parafollicular cells and is characterized by the secretion of calcitonin[5]. Unlike the most common papillary thyroid cancer (PTC), which has a 5-year survival rate of over 90%, 41–56% of patients with MTC had lymph node or distant metastases at the time of diagnosis, which could not be radically resected and responded poorly to chemotherapy and radiation, with a 10-year survival rate of only 40%[6–9]. Multi-target tyrosine kinase inhibitors (TKIs) vandetanib and cabozantinib had been suggested for advanced MTC by American Thyroid Association[10]. Although there is study found that TKIs could effectively improve patients' overall survival[11], recent research results indicate that both vandetanib and cabozantinib, as well as the new generation of highly selective TKI selpercatinib[12], can effectively enhance patients' progression-free survival, whether overall survival can be improved remains to be observed[13–15]. However, TKI treatment is accompanied by a significant high incidence rate of adverse reactions (38.9–72%), along with the common issue of resistance to long-term treatment, needing long-term follow-up and deeper mechanistic studies. Therefore, the investigation of novel therapeutic targets for MTC is of paramount importance.

CGRP, a transcriptional splicing product of calcitonin, has been found in the tumor and peripheral serum of MTC[16]. However, less is known about how CGRP affects the MTC microenvironment. Moreover, although the genomic and proteomic characteristics of MTC have been illustrated by a recent study[17], the understanding of its microenvironment remains largely unclear yet. Currently, research on the immune microenvironment of MTC is relatively limited, primarily focused on studies based on immunohistochemistry (IHC) or multiplex immunohistochemistry (mIHC) staining results, without consensus on the conclusions. A study in 2017 suggested low PD-1 positive staining and low immune infiltration in MTC[18]. However, a study published by Pozdeyev in 2020 using mIHC showed that 49% of primary lesions in MTC exhibited immune infiltration (mostly scattered or clustered around the tumor, with 14.6% cases observed within the tumor), which considered MTC was more immunologically than previous report[19]. Therefore, a deeper understanding of the immune microenvironment of MTC, including a comprehensive exploration of immune cell composition, gene expression, cellular status and cell-cell interactions[20], would provide direct human evidence on the impact of neurotransmitter CGRP on tumors and their microenvironment. More importantly, elucidating the underlying mechanism could also facilitate the discovery of potential therapeutic targets for MTC.

In this work, we conduct single-cell RNA sequencing analysis of tumors, adjacent normal thyroid tissues, and peripheral blood mononuclear cells (PBMC) from 7 patients with MTC and 8 with PTC, reporting an immunosuppressive microenvironment of MTC at single-cell resolution. Malignant cell-secreting CGRP can disrupt the development of intratumoral dendritic cells (DC) by hindering the downregulation of the negative regulator Kruppel Like Factor 2 (KLF2). The current study provides insights into the immunosuppressive microenvironment of MTC, and human evidence for the impact of neurotransmitter CGRP on tumors, proposing the CGRP receptor as a promising therapeutic target for MTC.

## Results

### Human thyroid cancer single-cell landscape reveals low immune infiltration in MTC

To generate a comprehensive transcriptional profile of the tumor microenvironment in human thyroid cancer, we performed single-cell RNA sequencing on the tumor tissue and paired adjacent normal thyroid tissue from 15 treatment-naive patients including 7 MTC and 8 PTC (Fig. 1A). Unsupervised clustering and uniform manifold approximation and projection (UMAP) analysis were performed on 228,400 cells from the tumor and adjacent normal tissue (Fig. 1B). Using the

characterized genes of each cluster including well-known annotation markers, we identified major cell types including T cells, B cells, plasma, myeloid cells, proliferative cells, follicular epithelial cells, parafollicular cells, fibroblasts, and endothelial cells. The differentially expressed genes of each major cell type are shown in Fig. 1C and Supplementary Data 1. We also analyzed cells isolated from 10 peripheral blood samples. Cell numbers, gene signatures, and counts of all samples are shown in Supplementary Table 1 and Supplementary Fig. 1A.

Thyroxine secretion genes like *TG*, *TSHR*, and *TPO* were used to identify follicular epithelial cells, which make up thyroid follicle structures and are the origin of PTC. (Supplementary Fig. 1B). On the other hand, parafollicular cells, the origin of MTC, were confirmed by calcitonin and neuroendocrine peptide such as gastrin releasing peptide (*GRP*) gene expression (Supplementary Fig. 1C)[21]. The annotations of tumor cells were confirmed by single-cell infer CNV analysis (Supplementary Fig. 1D, E). Furthermore, the Thyroid Differentiation Score (TDS) calculation verified the lower differentiation degree of PTC tumor cells than normal follicular epithelial cells, indicating that tumor cells underwent de-differentiation in tumor genesis as previously reported (Supplementary Fig. 1F)[22].

To generally profile the composition of the tumor immune microenvironment, we calculated the proportion of immune cells (T cells, B cells, plasma, myeloid cells, and proliferative cells) and non-immune cells, respectively. The proportion of immune cells was lower in MTC than that in PTC, as well as in public unsorted single-cell RNA dataset of other cancer types including PTC[23,24], anaplastic thyroid cancer (ATC)[24,25], breast cancer (BC)[26], gastric cancer (GC)[27], hepatocellular carcinoma (HCC), intrahepatic cholangiocarcinoma (ICC)[28] and known immunologically cold tumors including pancreatic cancer (PDAC)[29], glioblastoma (GBM)[30], and prostate cancer (PRAD)[31] (Fig. 1D and Supplementary Fig. 1G).

According to the enrichment ratio analysis of major cell types between PTC and MTC, all immune cells including T cells, B cells, plasma, and myeloid cells were found preferentially distributed in PTC, and basal cells showed relative enrichment in MTC (Fig. 1E). To validate the relatively lower immune infiltration in MTC, transcriptomic analysis also suggested that the immune infiltration score of MTC was significantly lower compared to PTC from our cohort or from The Cancer Genome Atlas Program (TCGA) cohort, indicating fewer immune cell distributions (Fig. 1F and Supplementary Fig. 1H). Using Immunohistochemistry (IHC) staining, fewer CD45[+] cells were found in the MTC tumor region as compared to PTC (Fig. 1G). In contrast to tumors, peripheral blood samples from PTC or MTC patients had similar compositions of immune cells, suggesting that distinct microenvironments observed in tumors were not ascribed to systemic immune disorders (Supplementary Fig. 1I, J). The single-cell atlas suggested that MTC could be recognized as an immune "cold" tumor.

### Tumor-specific CGRP interacts with DCs in MTC

Next, we wished to elucidate factors that "freeze" the microenvironment. First, differentiated gene and pathway enrichment analysis showed apparent enrichment of neuropeptide signaling pathway in MTC tumor cells and thyroid-stimulating hormone signaling pathway in PTC tumor cells, which may primarily stem from disparities in their respective cells of origin (Fig. 2A and Supplementary Data 2). Comparison tumor cells with their corresponding normal cells unveiled that PTC tumor cells exhibited significantly higher capacity for epithelial-mesenchymal transition (EMT), whereas MTC tumor cells displayed significant upregulation of pathways related to RNA splicing (Supplementary Fig. 2A, B). As expected, the gene *CALCA* encoding calcitonin and its alternative splicing product CGRP was highly expressed in MTC tumor cells (Fig. 2B). However, limited by 5-terminal sequencing, it was difficult to distinguish *CGRP* mRNA (exons 1-3 and 5-6) from calcitonin mRNA (exon 1–4) in our single-cell data[32]. To

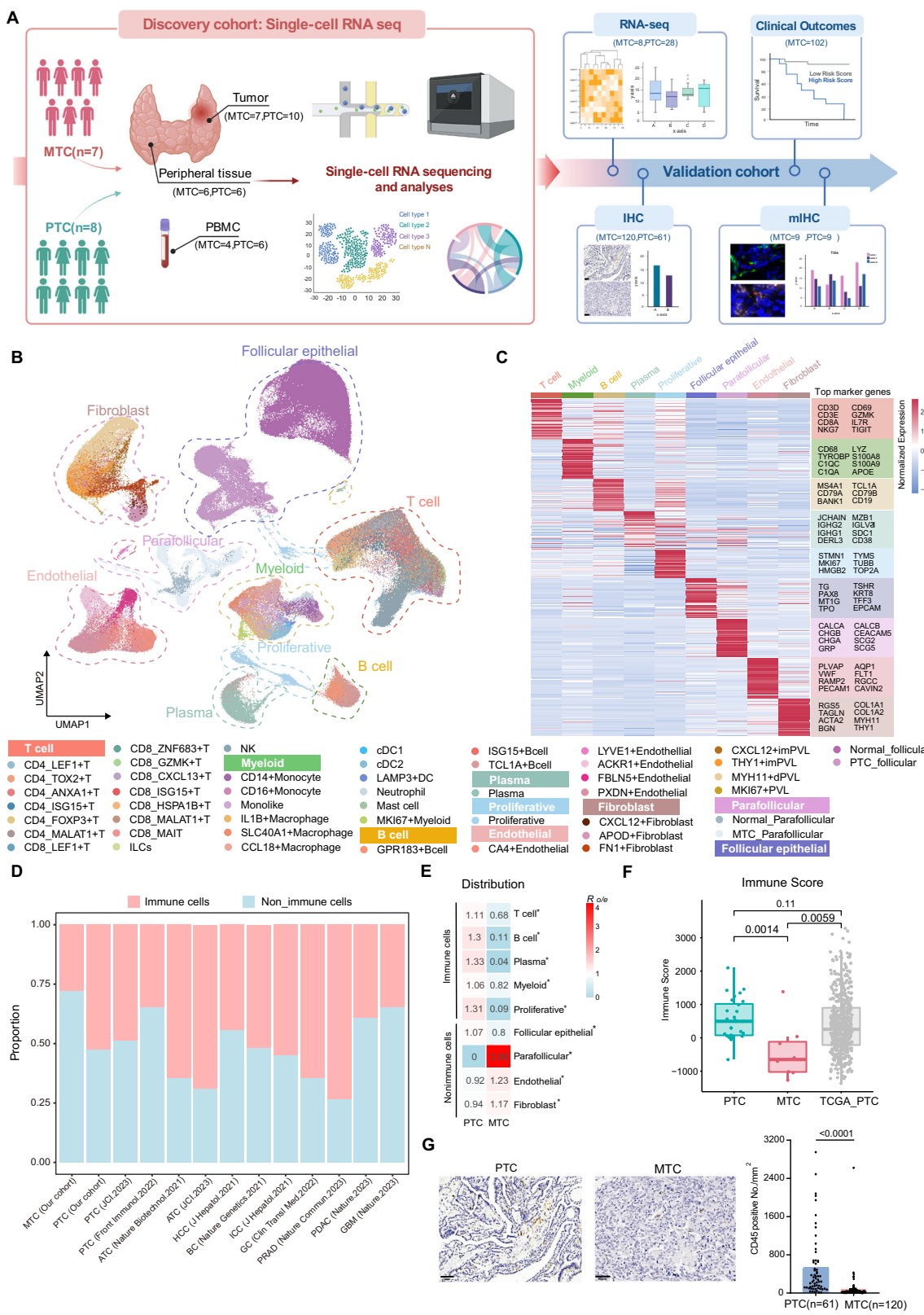

distinguish CGRP expression from calcitonin expression at the transcriptional level, bulk-RNA sequencing was enrolled, confirming the high expression of both *CGRP* mRNA and calcitonin mRNA in MTC (Fig. 2C). IHC also confirmed that CGRP and calcitonin were highly co-expressed at the protein level in the MTC (Fig. 2D). The expression of CGRP in MTC patients for single-cell RNA sequencing was shown in Supplementary Fig. 2C. Notably, MTC patients with

high intensity of CGRP expression had worse disease-free survival (DFS) ($p = 0.022$) with a hazard ratio of 2.775 (95% confident interval 1.116–6.901) (Fig. 2E).

Although high CGRP expression in MTC was associated with a poor prognosis, it is still unknown how CGRP produced by MTC tumor cells affects anti-tumor immunity. Recent studies suggested that CGRP could modulate tumor cells in oral mucosa carcinomas or T cells in

**Fig. 1 | Human thyroid cancer single-cell cell landscape reveals low immune infiltration in MTC. A** Workflow of our study. The left panel showed the experimental design of single-cell RNA-sequencing (scRNA-seq) as the discovery cohort. The right panel showed the type of validation experiments, including bulk-RNA sequencing, immunohistochemistry (IHC), and multi-immunohistochemical staining (mIHC), and their corresponding cohorts. Created with BioRender.com, released under a Creative Commons Attribution-NonCommercial-NoDerivs 4.0 International license. **B** Uniform manifold approximation and projection (UMAP) visualization of 228,400 cells from tumors and adjacent normal tissues, colored by cell type annotations. ILCs innate lymphoid cells, NK natural killer cell, DC dendritic cell, PVL perivascular cell, imPVL immature perivascular cell, dPVL differentiated perivascular cell. **C** Heatmap showing the expression patterns of marker genes in major cell types. **D** Cell type proportions of immune cells and non-immune cells. Each stacked bar represents one cancer type. PTC Papillary thyroid cancer, MTC Medullary thyroid cancer, ATC Anaplastic thyroid cancer, BC Breast cancer, GC Gastric cancer, HCC Hepatocellular carcinoma, ICC Intrahepatic cholangiocarcinoma, PRAD Prostate cancer, PDAC Pancreatic cancer, GBM Glioblastoma. **E** Heatmap showing the distribution ratio of major cell types in PTC and MTC. Calculated by Chi-square test, *$p < 0.01$. **F** Bar graph showing the predicted immune score of PTC (28 samples from our cohort and 502 samples from TCGA public dataset) and 8 samples MTC in the bulk-RNA data. The box plot illustrates the interquartile range in relation to the median, while the middle lines represent the median, and the lower and upper hinges denote the 25–75% interquartile range (IQR), with whiskers extending up to a maximum of 1.5 times IQR. *P*-value was determined using two-sided Wilcoxon rank-sum test. **G** IHC staining of CD45 in the tumor region of PTC and MTC (left panel). Dot plot shows the proportion of CD45[+] cells in PTC and MTC (right panel). *P*-values from two-tailed Student's *t* test were represented. Source data are provided as a Source Data file.

melanoma[2,3]. Cellchat was used to perform ligand-receptor analysis. Intriguingly, MTC tumor cells had a strong CALCA-CALCRL interaction with DCs but such interaction did not occur on either themselves or with T cells (Fig. 2F). As expected, no CALCA-CALCRL interaction was found in PTC, consistent with no CGRP expression in this tumor (Fig. 2F). Furthermore, among the receptor-ligand pairs between tumor cells and DC, the CALCA-CALCRL pair was the most significantly upregulated in MTC compared to PTC (Fig. 2G and Supplementary Fig. 2D, E). The receptor of CGRP is a complex of two proteins, Calcitonin receptor-like receptor (CALCRL) and Receptor activity modifying protein 1 (RAMP1)[33]. Meanwhile, only DCs express a relatively high level of both *CALCRL* and *RAMP1* (Fig. 2H, I). These results were further confirmed by multi-immunohistochemical staining. As shown in Fig. 2J, CGRP was highly expressed in MTC but not in PTC while 10–20% of intra-tumoral DCs expressed CALCRL in both PTC and MTC. Furthermore, previous studies have provided evidence that, as a G protein-coupled receptor, CGRP receptor activation can trigger downstream signalling pathways activation related to cyclic adenosine monophosphate (cAMP)[34–37]. By employing the Gene Ontology (GO) database for cAMP-related pathways as a gene signature, we observed a stronger activation of cAMP-related pathways in MTC DCs compared to PTC DCs (Supplementary Fig. 2F). Meanwhile, compared with DCs from public PTC, ATC dataset, a higher activation level of cAMP pathways of DCs in MTC was observed (Supplementary Fig. 2F). In in vitro cell experiments, we found that CGRP treatment induced an elevation in cAMP levels in DCs, consistent with previous reports (Supplementary Fig. 2G).

## DCs in MTC are dysfunctional and display distinct developmental trajectory compared to DCs in PTC

To further investigate the characteristics of tumor-infiltrating DCs in MTC, we defined transcriptional expression modules of all DCs using consensus non-negative matrix factorization (cNMF) and graph clustering-based approach at the sample level. DCs from all tissue samples were divided into distinct gene-expression programs, which clustered into specific modules based on their expression similarity (Fig. 3A). Fewer specific gene expression programs were formed in MTC compared to PTC, when NMF clustering was performed separately by tumor type (Supplementary Fig. 3A–C). Further pathway enrichment analysis of the characterized gene signatures revealed specific immune functions of each DC module 1-6. For example, DCs in module 1 showed increased activity in immune cell activation, while those in modules 4 and 6 showed specific responses to interferon and TNF-alpha respectively. Response to the bacteria pathway was more active in cells from module 2, and the virus-related pattern was enriched in module 3. Particularly, DCs without a specific immune-related function were gathered and defined as the "Other" group. Analysis of tissue distribution of DC modules showed MTC-infiltrated DCs were decreased in immune-functional modules 1-6, while strongly increased in the "Other" group (Fig. 3B and Supplementary Fig. 3A). A correlation analysis between functional scores and the top 20 genes signature of each DC module was performed. As predicted, the signature of cells in the "Other" group showed the strongest negative correlation with both DC antigen-presenting and co-stimulatory function (Fig. 3C).

As indicated by cNMF analysis, the gene score of antigen-presentation and co-stimulation, and the expression of classical co-stimulatory factor were both downregulated in MTC DCs compared with PTC (Fig. 3D and Supplementary Fig. 3D). When compared with adjacent normal tissue respectively, DCs of MTC had a stronger antigen-presenting function but attenuated co-stimulatory ability, but both of functions were stronger in DCs of PTC (Supplementary Fig. 3E). At the same time, the expression of classical co-stimulatory genes *CD80*, *CD86* and *CD40* was lower in DCs in MTC, confirmed by bulk-RNA data (Supplementary Fig. 3F). Low levels of HLA-DR and CD86 in MTC was also validated by multi-immunohistochemical staining (Fig. 3E).

Intra-tumoral DCs originate from circulating precursors of conventional DCs or monocytes[38]. Two potential DC developmental trajectories were discovered by the pseudo-time analysis when CD14[+] monocytes were used as an early-stage reference. (Fig. 3F and Supplementary Fig. 3G). In more detail, cells in trajectory 2 were dominated by PTC-derived DCs and differentiated toward an increasing costimulatory function, while cells in trajectory 1 enriched for MTC-derived DCs, differentiated in the opposite direction (Fig. 3F, G). The cells in the trajectory can be classified into three distinct states. State 1 corresponds to the dysfunctional dendritic cells (DCs) at the end of trajectory 1, State 2 corresponds to the functional DCs at the end of trajectory 2, and State 3 is primarily composed of shared monocytes (Fig. 3H and Supplementary Fig. 3G). When integrating the cell trajectory with the state information, trajectory 1, displayed a milder increase in *CD40*, *CD80* and *CD83* compared to trajectory 2 (Fig. 3I and Supplementary Fig. 3H). When comparing the pseudo-time data of DCs in MTC or PTC, similar kinetics of costimulatory factor expression were discovered, reflecting the fact that PTC and MTC DCs were enriched in trajectories 1 and 2 with different functional states, respectively (Supplementary Fig. 3I).

Based on the findings of impaired differentiation of DCs in MTC, we scored all cells in the microenvironment according to the negative regulation of DC differentiation signature from GO dataset. Macrophages scored the highest, followed by MTC tumor cells in MTC (Supplementary Fig. 3J, K). Comparison of MTC and PTC tumor cells revealed significantly higher negative regulation scores of MTC tumor cells (Supplementary Fig. 3L). Furthermore, there was a significant negative correlation between negative regulation scores in tumors and DC co-stimulation scores (Supplementary Fig. 3M). These results indicated the impaired development and co-stimulatory function of DCs in MTC and also supported that the CGRP-secreted MTC tumor cells are likely an important factor contributing to DC differentiation impairment.

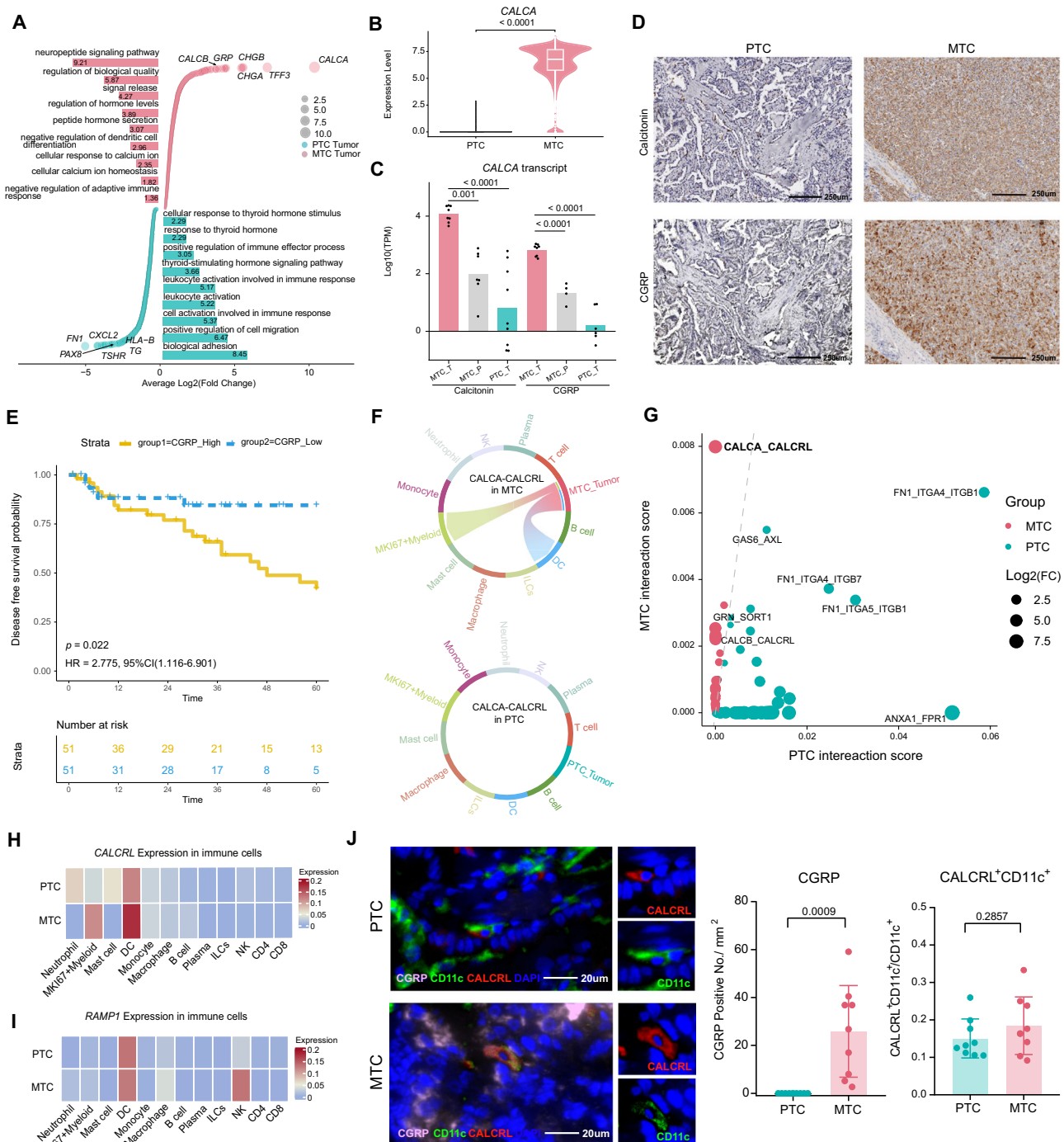

**Fig. 2 | Tumor-specific CGRP interacts with DCs in MTC. A** Differentially expressed genes and enrichment pathways of malignant cells between PTC and MTC were shown. Dot size and bar length represent the average Log2 (fold change) of genes and pathways, respectively. **B** Violin plot showing the high expression of *CALCA* in MTC tumor cells. The box inside illustrates the interquartile range in relation to the median, while the middle lines represent the median, and the lower and upper hinges denote the 25–75% interquartile range (IQR), with whiskers extending up to a maximum of 1.5 times IQR. Calculated by two-sided Wilcoxon rank-sum test. **C** In the bulk-RNA data, the *CALCA* transcripts and *CGRP* transcripts were counted by TPM in tumor and peripheral tissue of PTC and MTC and plotted as bars. MTC_T ($n = 8$) and PTC_T ($n = 9$ in calcitonin and $n = 6$ in CGRP) represents tumors of MTC and PTC, respectively. MTC_P ($n = 7$ in calcitonin and $n = 4$ in CGRP) represents peripheral thyroid tissue of MTC. *P*-value was determined using one-way ANNOVAR test. **D** In the tumor region of PTC and MTC, IHC staining for calcitonin and CGRP was shown. These images are representative image from PTC group

($n = 3$) and MTC group ($n = 3$). **E** Kaplan–Meier curve showing the disease-free survival (DFS) of MTC patients grouped by the intensity of CGRP expression in the tumor region. MTC patients with high CGRP expression were characterized by yellow color while patients with low expression were characterized by blue color. *p*-value was calculated by Log-rank test. **F** CALCA-CALCRL ligand-receptor interaction between tumor cell and all immune cell types was shown by chord plot in MTC (top panel) and PTC (bottom panel). **G** Scatter plot showing cell-cell interaction pairs between tumor cell and DC. The dotted line indicates the same interaction score. **H** Heatmap showing the expression of *CALCRL* in all types of immune cells. **I** Heatmap showing the expression of *RAMP1* in all types of immune cells. **J** mIHC showing the expression of CGRP, the percentage of CALCRL- positive CD11c+ cells in PTC ($n = 9$) and MTC ($n = 9$) tumor regions. Data were expressed as the mean ± SD. *P*-values from two-tailed Student's *t* test were represented. Source data are provided as a Source Data file.

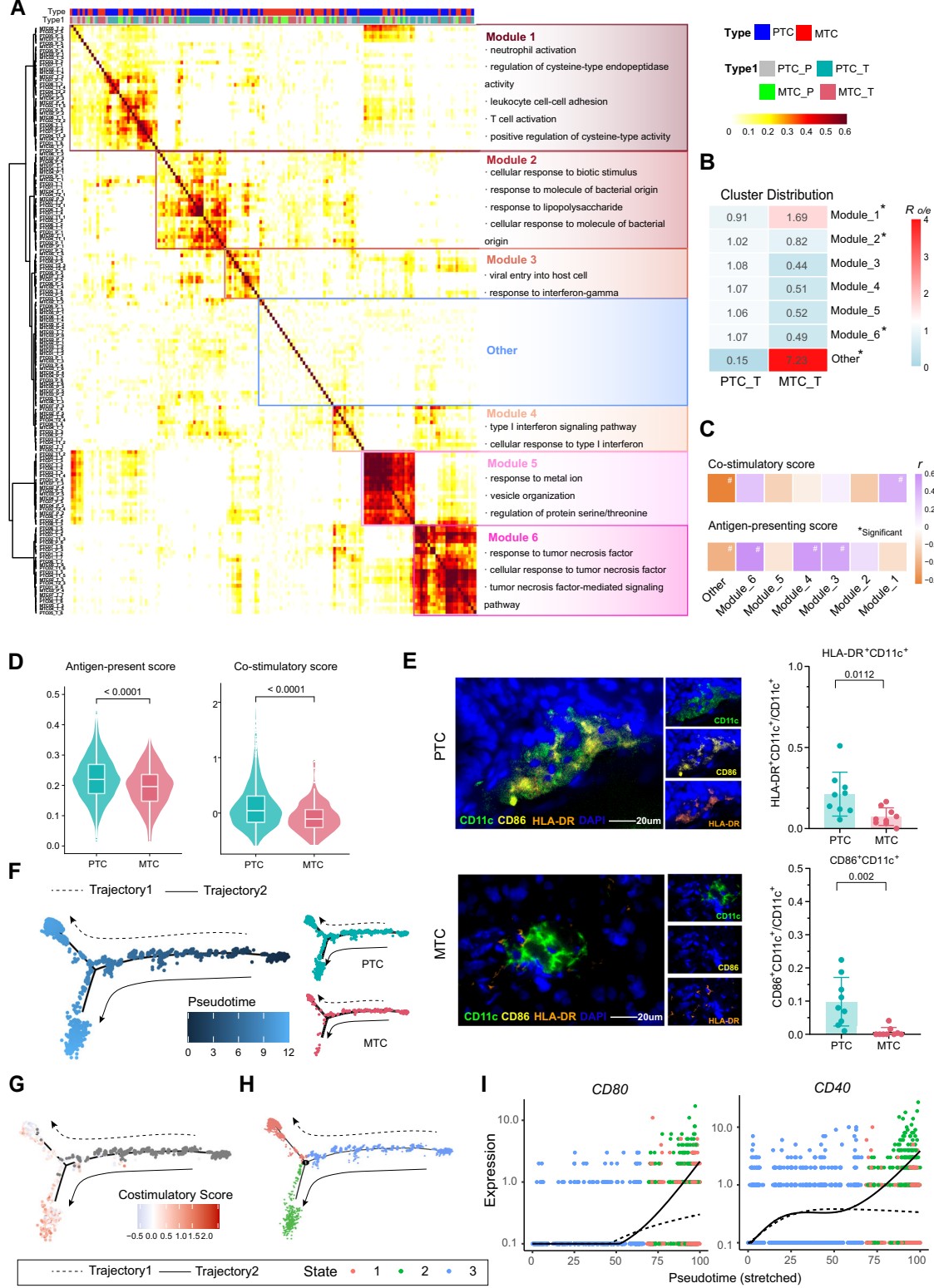

**Elevating level of transcription factor KLF2 contributes to the development of intra-tumoral DCs in MTC**

To gain insights into the regulatory mechanisms underlying DC development, we employed the CellOracle strategy to analyze the changes in transcription factor (TF) between different functional states of DCs[39]. Based on the dimension calculated by Monocle, CellOracle successfully validated the developmental trend from monocytes to dendritic cells (DCs) under current conditions (Fig. 4A). Network analysis was performed to identify the top 30 key transcription factors

for DCs of different states or tissue origins. The analysis showed the metrics of degree centrality, betweenness centrality and eigenvector centrality to determine the importance of each transcription factor (Supplementary Fig. 4A). The key transcription factors for each group were identified by considering the overlapped transcription factors in degree centrality, betweenness centrality and eigenvector centrality. Further screening revealed that both Kruppel-like factor 2 (*KLF2*) and *JUN* met the criteria of being specific and key transcription factors in both State1 and MTC-derived DCs (Supplementary Fig. 4B). Expression

**Fig. 3 | DCs in MTC are dysfunctional and display distinct developmental trajectory compared to DCs in PTC. A** Heatmap showing expression correlation of gene programs in DCs across tumor and normal tissue samples from PTC and MTC patients and clustered by column. Based on the correlated-expression relationship, modules consisting of gene programs were defined and characterized by immune pathways (enriched by the top gene signature of each module). Here, PTC and MTC represent tumors of PTC and MTC respectively, as above. PTC_P represents normal thyroid tissue of PTC and MTC_P represents normal thyroid tissue of MTC. **B** Heatmap showing the distribution ratio of DC modules in PTC and MTC. Calculated by Chi-square test, \*p < 0.01. **C** Heatmap showing the correlation coefficient between module gene signature scores and DC antigen-presenting and co-stimulatory scores. Calculated by Spearman or Pearson's correlation test, #p < 0.05. **D** The gene scores of antigen-presenting, co-stimulatory signatures were calculated in DCs between PTC and MTC and shown in violin plots. The box inside illustrates

the interquartile range in relation to the median, while the middle lines represent the median, and the lower and upper hinges denote the 25-75% interquartile range (IQR), with whiskers extending up to a maximum of 1.5 times IQR. Calculated by two-sided Wilcoxon rank-sum test. **E** Multiplex immunofluorescence (mIHC) image showing HLA-DR and CD86 expression on CD11c$^+$ cells in PTC (n = 9) and MTC (n = 9) tumor regions (left panel). Bar graph shows the proportion of HLA-DR$^+$ and CD86$^+$ DCs in PTC and MTC calculated by two-sided Student's t-test. **F** Left panel shows the pseudotime trajectory of CD14$^+$ monocytes and DCs derived from Monocle2. The right panel showing DCs from PTC and MTC along the trajectory, respectively. **G** The trajectory of DCs was colored by the score of the co-stimulatory gene signature. Monocytes are shown in gray without score calculation. **H** The trajectory was colored by cell state. **I** Pseudo-time expression of the genes CD80 and CD40 along the trajectories 1 and 2. Source data are provided as a Source Data file.

analysis revealed that KLF2, with higher average ranking than JUN, decreased quickly over time in trajectory 2 but at a slower pace in trajectory 1 (Fig. 4B, C). Performing a knockout (KO) simulation of KLF2 using CellOracle, we showed the changes in cell developmental trajectories after the knockout in Fig. 4D. The perturbation score is used to evaluate how KLF2 KO affects the directionality of cell differentiation. A negative score (in purple color) indicates that KLF2 KO delays or blocks differentiation, while a positive score (in green color) suggests promotion of differentiation. Perturbation score analysis predicted that KLF2 KO in the intersection of State1, State2, and State3 inhibited differentiation from monocytes to DCs, highlighting the crucial role of KLF2 in DC early-differentiation (Fig. 4E). Notably, in further differentiation of DCs, KLF2 KO also leads to a significant promotion of differentiation towards State2 cells, while the effect of differentiation towards State1 cells is attenuated or even blocked, suggesting KLF2's critical role in differentiation between State1 and State2 (Fig. 4E). Furthermore, in the cNMF analysis mentioned earlier, we also observed that KLF2 was the only shared specific marker gene between the "Other" group and MTC DCs (Supplementary Fig. 4C–E).

KLF2, a member of the Kruppel family of transcription factors, is a regulator of cellular activation. Monocytes, T cells, and B cells exhibit high levels of KLF2 expression when they are in their early developmental or resting stage[40]. In contrast, well-developed DCs express a low level of KLF2[41]. When DCs were divided into KLF2$^+$ or KLF2$^-$ groups based on the presence or absence of KLF2 expression, KLF2$^+$ DCs were more enriched in MTC, with a fraction of about 75% (Fig. 4F and Supplementary Fig. 4F, G). Within MTC, KLF2$^+$ DCs were less functional than KLF2$^-$ DCs (Fig. 4G). Interestingly, we also found that KLF2 expression was significantly higher in MTC DCs and that this expression had a negative correlation with co-stimulatory function (Fig. 4H, I and Supplementary Fig. 4H). Furthermore, we integrated public dataset and compared the co-stimulatory scores as well as the expression levels of KLF2 of DCs among various thyroid tumors. The results indicated that in MTC, PTC, and ATC tumor tissues, DCs exhibited the lowest co-stimulatory function scores in MTC and the highest expression levels of KLF2 (Supplementary Fig. 4I, J). A negative correlation was also observed between KLF2 expression and co-stimulatory score (Supplementary Fig. 4K). What's more, along the developmental trajectory from monocyte to DCs, KLF2 expression was decreased quickly in PTC DCs, but maintained at a relatively high level in MTC DCs over time (Fig. 4J). These data suggested that the dynamic change of KLF2 might contribute to the development of intra-tumoral DCs in MTC.

### CGRP drives the development of dysfunctional DCs by preventing the loss of KLF2
Given both CGRP and KLF2 were associated with the dysfunctional DCs in MTC, we next asked if CGRP regulated KLF2 expression. In the context of CGRP activating the cAMP pathway, a positive correlation was observed between the cAMP activation score and KLF2 expression

in tumor DCs, indicating the potential relationship between changes in KLF2 expression and CGRP receptor activation (Supplementary Fig. 5A). Further integration of our cohort with publicly available single-cell RNA data revealed that in tumor, the activation level of the cAMP pathway in DCs is positively correlated with the expression level of KLF2, while it is negatively correlated with co-stimulatory function (Supplementary Fig. 5B-C).

To further confirm the relationship between CGRP, cAMP pathway and KLF2 in DCs, monocytes were isolated from PBMCs and cultured with GM-CSF and IL-4 to drive the differentiation of these precursors into immature DCs. Recombinant human CGRP was added along with the differentiation process to mimic the persistence of CGRP stimulus in MTC (Fig. 5A). As mentioned earlier, CGRP effectively induces an increase in cAMP concentration in DCs (Supplementary Fig. 2G). As what had been found in single-cell data, KLF2 expression was dramatically decreased during the in vitro differentiation and maturation (Fig. 5B-C and Supplementary Fig. 5D). Interestingly, treatment with CGRP slowed the rapid loss of KLF2 (Fig. 5B). Although CGRP treatment did not alter the expression of costimulatory factors in immature DCs (Supplementary Fig. 5E), the maturation of these DCs in response to stimuli was significantly impaired (Fig. 5D and Supplementary Fig. 5F). In detail, as compared to those developed in the absence of CGRP, DCs that were exposed to CGRP during differentiation had significantly lower levels of CD40, CD80, CD83, CD86, and HLA-DR expression after cytokine stimulation. Encouragingly, Rimegepant, a small-molecule CGRP receptor antagonist used to treat migraines, and SQ22536, an adenylate cyclase inhibitor which effectively inhibits the production of intracellular cAMP and suppresses the activation of the cAMP pathway, could reduce the elevated levels of KLF2 induced by CGRP and offset CGRP's detrimental impact on DC development (Fig. 5C, D and Supplementary Fig. 5F).

Collectively, these results revealed that CGRP secreted by tumor cells could drive an abnormal development of intra-tumoral DCs by cAMP pathway activation and preventing the loss of KLF2. CGRP receptor could be a potential therapeutic target for MTC, in which CGRP receptor antagonists might be able to restore functional DCs.

### The inactivated status of CD8$^+$ T cells in MTC
Because DCs are the key antigen-presenting cells to induce T cell responses, we next explored the status of anti-tumor T cell responses in MTC when DCs were incompetent. Compared to PTC, MTC had a much lower proportion of T cells (Fig. 1E). To comprehensively dissect the influence on tumor-infiltrating T cells caused by dysfunctional DCs, we re-clustered and investigated T cells, the largest cell type of immune cells as well as a remarkable target of immunotherapy (Fig. 6A, Supplementary Fig. 6A and Supplementary Data 3). Differentially expressed gene analysis of CD8$^+$ T cells, CD4$^+$ T cells, NK cells and ILCs between MTC and PTC, showed that CD8$^+$ T cells had the most pronounced transcriptional alteration (Fig. 6B and Supplementary Data 4).

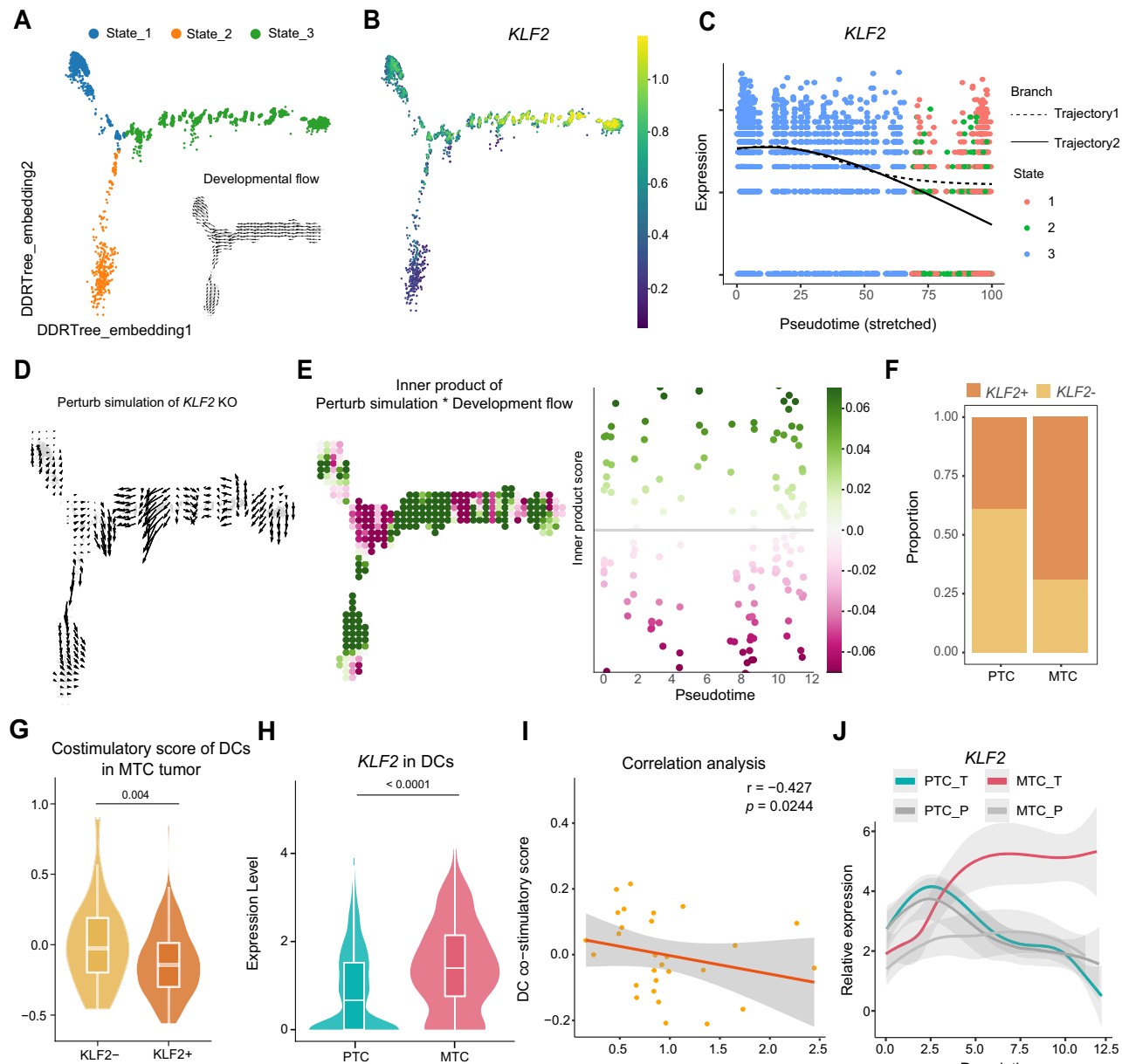

**Fig. 4 | Elevating level of transcription factor KLF2 contributes to the development of intra-tumoral DCs in MTC. A** The Monocle-based trajectory analysis coordinates were utilized to import state information into CellOracle for further analysis. Re-calculated of pseudotime trajectory and converted into a 2D pseudo-time gradient vector. **B** Expression of *KLF2* projected onto trajectory. **C** Pseudo-time expression of *KLF2* along trajectories 1 and 2. **D** CellOracle *KLF2* knockout simulations showing cell-state transition vectors along trajectory. **E** *KLF2* KO simulation with perturbation scores calculated according to the change in vector direction after knockout compared to the original vector direction. A negative score (in purple color) indicates that TF knockout delays or blocks differentiation, while a positive score (in green color) suggests promotion of differentiation. **F** Cell type proportions of *KLF2*+ and *KLF2*- DCs in MTC and PTC. **G** Violin plot showing the co-stimulatory score of *KLF2*+ and *KLF2*- DCs in MTC. The box inside illustrates the interquartile range in relation to the median, while the middle lines represent the median, and the lower and upper hinges denote the 25–75% interquartile range (IQR), with whiskers extending up to a maximum of 1.5 times IQR. Calculated by two-sided Wilcoxon rank-sum test. **H** Violin plot showing the *KLF2* expression in PTC and MTC DCs. The box inside illustrates the interquartile range in relation to the median, while the middle lines represent the median, and the lower and upper hinges denote the 25–75% interquartile range (IQR), with whiskers extending up to a maximum of 1.5 times IQR. Calculated by two-sided Wilcoxon rank-sum test. **I** Scatter plot showing the correlation relationship between *KLF2* expression and the co-stimulatory score of DCs at the sample level (*n* = 28). *r* indicates the correlation coefficient calculated by Spearman correlation test. **J** Plots showing the expression of *KLF2* in the tumor or in the peripheral normal tissue of PTC and MTC along the trajectory. MTC_T and PTC_T represents tumors of MTC and PTC, respectively. MTC_P and PTC_P represents peripheral thyroid tissue of MTC and PTC, respectively. Source data are provided as a Source Data file.

Meanwhile, as shown in Supplementary Fig. 6B, C, differentially expressed genes (DEGs) and gene set enrichment analysis both showed downregulated expression of cytokines such as granzymes *GZMK*, *GZMA* and *NKG7* and cytotoxic pathways activity in MTC. In addition, CD4+ T cells in MTC downregulated T cell activation pathways (Supplementary Fig. 6D, E).

In the further functional analysis of CD8+ T cells, significantly higher naive-like gene scores, lower scores and expression of cytotoxic and dysfunctional genes were observed in MTC as compared to PTC as well as paired adjacent normal tissues (Fig. 6C, D and Supplementary Fig. 6F–H). Furthermore, we integrated public single-cell dataset to analyze the functional status of CD8+ T cells in MTC, PTC, and ATC, as

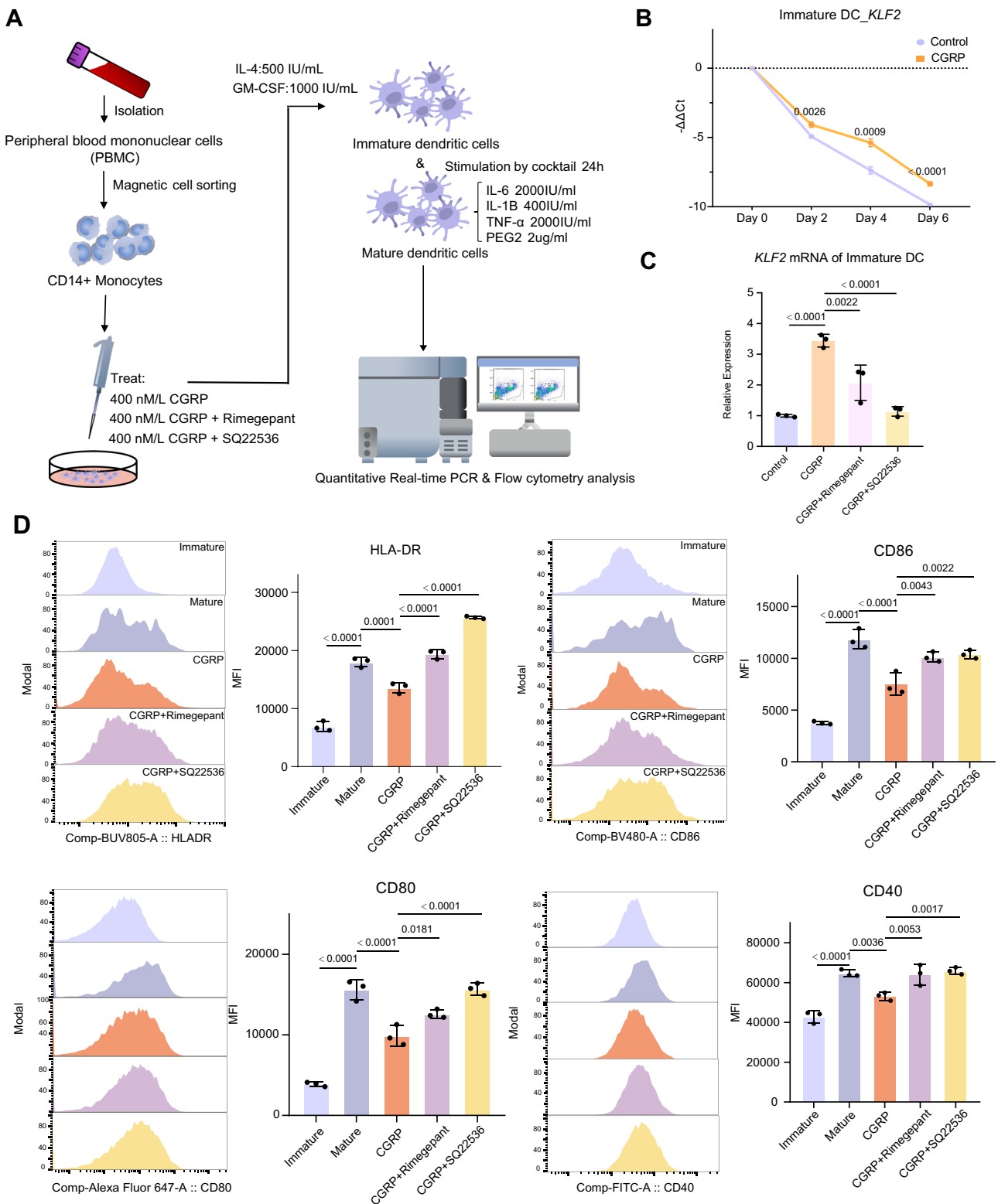

**Fig. 5 | CGRP drives the development of dysfunctional DCs by preventing the loss of KLF2. A** Workflow of in vitro experiments. Monocytes isolated from PBMC were cultured and induced into immature DCs by specific cytokines. During induction, DCs were treated with 400 nM CGRP. After inducing maturation by cytokine cocktail for 24 h, DCs were harvested for RNA extraction and flow cytometry analysis. **B** Real-time PCR analysis of *KLF2* transcripts at day 0,2,4 and 6 during DC induction (*n* = 3 for each timepoint). The box illustrates the interquartile range in relation to the median, while the middle lines represent the median, and the lower and upper hinges denote the 25–75% interquartile range (IQR), with whiskers extending up to a maximum of 1.5 times IQR. *P* value between groups in

one day was calculated by unpaired student's *t* test. **C** Real-time PCR analysis of *KLF2* transcripts at day 6 during DC in different groups (*n* = 3 for each group). The data was presented as mean ± Standard deviation (SD). *P* values between groups were calculated by one-way ANNOVAR. **D** Representative flow cytometry histogram and increasing degrees of mean fluorescence intensity (MFI) of co-stimulatory markers CD80, CD40, CD86 and HLA-DR expressed on DCs (*n* = 3 for each group). *P* values between groups were calculated by one-way ANNOVAR. The data was presented as mean ± SD. All experiments were performed independently for three times. Source data are provided as a Source Data file.

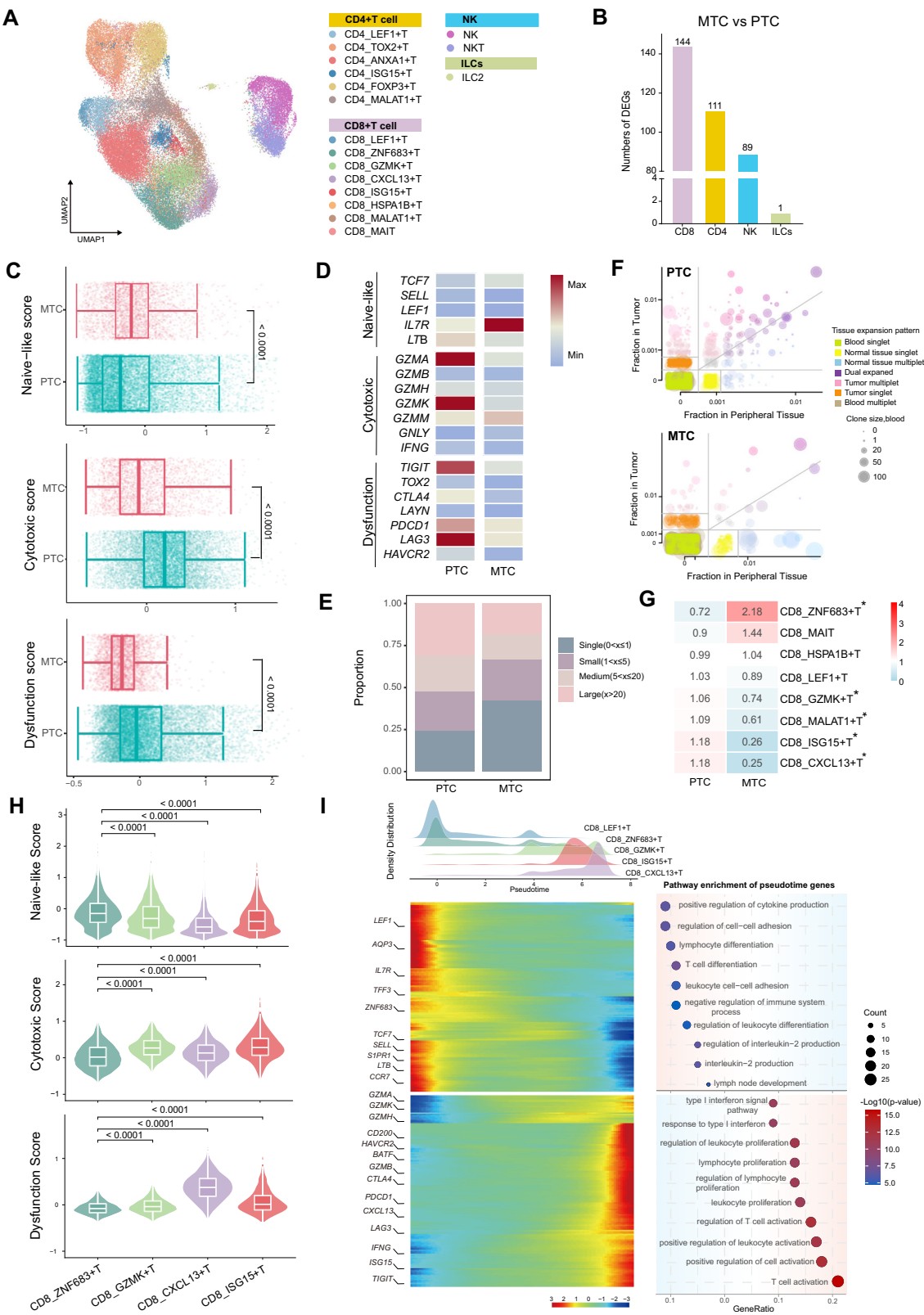

depicted in Supplementary Fig. 6I, J. The MTC group exhibited the lowest scores for cytotoxicity and dysfunction. Transcriptomic data similarly indicated that compared to the PTC cohort from our cohort or from the TCGA database, MTC displayed lower levels of cytotoxic and exhaustion functions (Supplementary Fig. 6K–M). In addition, further investigation of CD8⁺ T cells with cytotoxicity, TCR clonal expansion analysis showed that the proportion of largely (> 20) and

moderately expanded (5 < x ≤ 20) CD8⁺ T cells decreased in MTC (Fig. 6E and Supplementary Fig. 6N). It has been recognized that T cell clones derived from peripheral blood or presenting dual-expanded in tumor and peripheral normal tissues may represent a stronger immune response[42]. When performing shared TCR analysis in tumor, peripheral normal tissues and PBMC from patients with integral samples[43], MTC has fewer dual-expanded TCR clones in tumor and

**Fig. 6 | The inactivation of CD8[+] T cells in MTC. A** Re-clustering of T cells is shown in the UMAP plot annotated with subpopulations. **B** Number of differentially expressed genes (DEGs) between MTC- and PTC- derived CD8[+]T cells, CD4[+]T cells, NK cells and ILCs. **C** Comparison of the scores for naive-like, cytotoxic and dysfunctional gene expression of CD8[+] T cells between MTC and PTC. The boxplot illustrates the interquartile range in relation to the median, while the middle lines represent the median, and the lower and upper hinges denote the 25−75% interquartile range (IQR), with whiskers extending up to a maximum of 1.5 times IQR. Calculated by two-sided Wilcoxon rank-sum test. **D** Dot plot showing the expression of naive-like, cytotoxic and dysfunctional genes in CD8[+] T cells of PTC and MTC. **E** Bar graph showing the proportions of T cell receptor (TCR) expansion levels in MTC and PTC. *X* represents the clonal size of the TCR clonotypes. **F** Scatter plot of cross-tissue clonal expansion analysis was shown for patients with tumor sample, adjacent normal thyroid tissue sample and blood sample. Dots were sized for blood clonal size and colored according to tissue expansion pattern. Equal cell

proportions were indicated by diagonal lines, and the absence or presence of clones within compartments were separated by other lines. **G** Heatmap showing the distribution ratio of CD8[+] T cell subpopulations in PTC and MTC, *p < 0.01.* **H** Violin plot showing the gene scores of naive-like, cytotoxic and dysfunctional in differentially distributed subpopulations of CD8[+] T cells. The box inside illustrates the interquartile range in relation to the median, while the middle lines represent the median, and the lower and upper hinges denote the 25−75% interquartile range (IQR), with whiskers extending up to a maximum of 1.5 times IQR. Calculated by two-sided Wilcoxon rank-sum test. **I** Density plot showed the distribution of CD8[+] T cell subpopulations along the pseudo time trajectory (upper-left panel). Heatmap showed the changing gene expression over time (lower-left panel). Pathway enrichment analysis was displayed in a bubble plot showing the differential pathway activity of genes located at the beginning and end of the trajectory, respectively. The dot size represented gene counts and the color represented -Log10(*p*-value) (lower-right panel).

normal tissues as well as fewer parallel expanded TCR clones in blood than PTC, indicating an attenuated immune response and chemotaxis (Fig. 6F).

As shown in Fig. 6G, CD8[+] T cell ratio analysis displayed a strong distribution preference of ZNF683[+]T cells in MTC and a significant enrichment of GZMK[+] T cells, MALAT1[+] T cells, ISG15[+] T cells and CXCL13[+] T cells in PTC. Functional analysis revealed that ZNF683[+] T cells were characterized by ZNF683 (marker of long-lived memory progenitors) and HOPX (prominent regulator of early differentiation of naive T cells) with high naive-like score[44,45]. GZMK[+] T cells had the highest cytotoxic score, while CXCL13[+] T cells featured by immune checkpoint genes had the highest dysfunctional score (Fig. 6H and Supplementary Fig. 6A). MALAT1[+] T cells were considered low-quality and excluded from analysis due to low gene median. We also performed pseudo-time ordering analysis of CD8[+] T cells using the Monocle2 algorithm (Supplementary Fig. 6O). LEF1[+] T cells and ZNF683[+] T cells were distributed mainly at the beginning of the pseudo-path, while others were distributed at the end (Fig. 6I and Supplementary Fig. 6P). Consistent with the trajectory from naive to cytotoxic and further dysfunctional status (Supplementary Fig. 6Q, R), MTC CD8[+] T cells clustered mainly at the beginning of the trajectory when we grouped cells by tumor type (Supplementary Fig. 6S). Pseudo-time gene distribution and pathway enrichment analysis showed that the expression of naive genes and T cell differentiation pathways, weakened along with the trajectory. In contrast, cytotoxicity and immune checkpoint genes, showing enrichment of T cell activation and proliferation pathways, increased over time (Fig. 6I). Taken together, these results revealed an immunosuppressive microenvironment in MTC characterized by a less active status of CD8[+] T cells.

### Dysfunctional DCs influenced by CGRP are responsible for inducing the suppressive characteristics of CD8[+] T cells in MTC

In consideration of the previous study suggesting a potential direct effect of CGRP on T cells[2], we conducted in vitro experiments involving CGRP and CD8[+]T cells to explore whether the characteristics of CD8[+] T cells in the MTC microenvironment are directly influenced by CGRP. CD8[+] T cells were isolated from PBMCs and cultured with CD3/CD8 beads and IL-2 for 4 days, with CGRP or CGRP combined CGRP receptor inhibitor Rimegepant in the treatment groups. Subsequently, Carboxyfluorescein succinimidyl ester (CFSE) proliferation assays and flow cytometry for CD8[+]T cell cytotoxicity molecules were performed. The results revealed that long-term exposure to CGRP did not affect T cell proliferation or its expression of functional molecules (Supplementary Fig. 7A, B).

Macrophages also constitute an important population of myeloid cells with antigen presentation and immunomodulatory roles other than DCs. Further analyses were performed and characteristic analysis based on subtypes showed that IL-1B[+]Macrophages had the highest M1

(pro-inflammatory subtype) score, while CCL18[+]Macrophages had the highest M2 (anti-inflammatory subtype) characteristics scores (Supplementary Fig. 7C, D). Referring to previously published studies on the expression of immunosuppressive gene signature in macrophages[46,47], the results suggested that CCL18[+]Macrophages might be a subtype with certain immunosuppressive functions, as their scores for immunosuppression-related genes were the highest among the three subtypes (Supplementary Fig. 7E). Further analysis of the distribution between MTC and PTC revealed that all three subtypes of macrophages were relatively more abundant in PTC tumors (Supplementary Fig. 7F). Scoring analysis of macrophages based on tissue origin of MTC and PTC showed that both M1 and M2 scores of macrophages in MTC were slightly lower than those in PTC (Supplementary Fig. 7G, H). Comparative analysis showed no significant difference in immunosuppression-related gene scores and genes was observed in MTC macrophages (Supplementary Fig. 7I, J). The lack of significant differences in macrophages between the two types of tumors suggested that macrophages may not be the primary cell population leading to significantly different immune-responses in MTC and PTC.

Based on the results of correlation analysis, DCs were the only cell population showing a both positive correlation with the activation level of CD8[+] T cells in terms of cell proportion, antigen presentation, and co-stimulatory function (Supplementary Fig. 7K–M). In interaction analysis, a primary but attenuated co-stimulatory interactions of the DC-T cell axis in MTC compared to PTC, suggesting the dysfunctional DCs may be responsible for attenuated T cell responses (Fig. 7A). Considering the inability to accurately locate the expression of *CGRP* at the single-cell transcriptome level, we analyzed the relationship between *CGRP* expression and the immunosuppressive microenvironment characteristics of MTC based on bulk-RNA data. The results suggested that the expression level of *CGRP* in tumors was negatively correlated with the immune infiltration score, the cytotoxicity score, the exhaustion score of CD8[+] T cells, and the co-stimulatory function score of DCs in the immune microenvironment (Fig. 7B). In summary, we revealed the crucial role of dysfunctional DCs influenced by CGRP in shaping the immune suppressive microenvironment of MTC.

## Discussion

Recent studies have revealed a tumor-promoting function of neuropeptides. However, key evidence is limited and mainly from in vitro and animal studies to date. In the present study, we mapped a comprehensive single-cell landscape of MTC, revealing an immunosuppressive microenvironment. Additionally, we demonstrated in humans that CGRP could disrupt the development of intra-tumoral DCs, and reported a mechanism by which CGRP prevented the downregulation of the negative regulator KLF2 during DC development (Fig. 7C).

The nervous and immune systems have been studied independently for a long time, and recent advances in neuroimmunology have

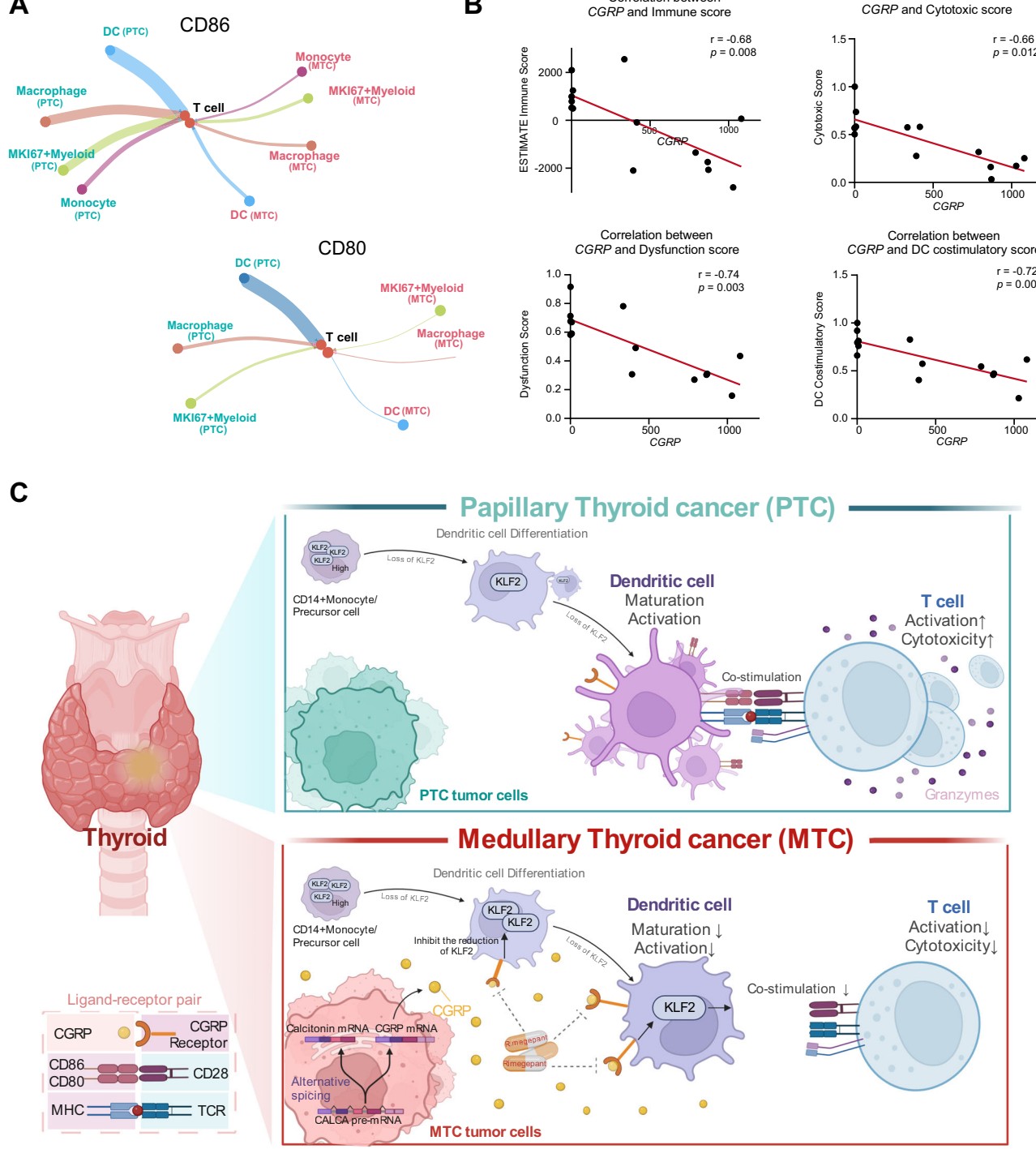

**Fig. 7 | Dysfunctional DCs influenced by CGRP responsible for inducing the suppressive characteristics of CD8⁺ T cells in MTC. A** Circle plot showing the cell-cell interaction strength of CD86 (left panel) and CD80 (right panel) between T cells and other immune cell types in PTC (green) and MTC (red). **B** Scatter plot showing the correlation relationship between *CGRP* expression and the Immune score, the cytotoxic score of CD8⁺T cell, the dysfunctional score of CD8⁺T cell and the co-stimulatory score of DCs in tumor bulk-RNA data (*n* = 14). r indicates the correlation coefficient calculated by Spearman correlation test. **C** Schematic representation of CGRP-induced immunosuppressive tumor microenvironment in MTC. Source data are provided as a Source Data file. Created with BioRender.com, released under a Creative Commons Attribution-NonCommercial-NoDerivs 4.0 International license.

revealed the communication and connections between the two systems. Investigating relationships between nerves and immune cells, studies are increasingly focusing more on neuropeptides secreted by nociceptive nerves, such as CGRP, a mediator in neuroimmunity[48]. For example, in skin immunity, the function of epidermal DCs could be suppressed by CGRP released by sensory neurons[49,50]. In allergic inflammation, the production of cytokines and cell proliferation of type 2 innate lymphoid cells (ILC2) could be inhibited by neuron-derived CGRP[51]. However, again, key evidence was mainly from in vitro and animal studies and the working mechanism underlying how CGRP modulates DC functions in humans remains largely unclear. The lack of a proper model is a major obstacle, since neuron-secreting CGRP mostly participates in chronic inflammatory diseases of the skin, respiratory system or gastrointestinal tract, in which obtaining lesions

and healthy tissue controls may be difficult in humans due to ethical issues.

Alternatively, CGRP is not only secreted by nociceptors, and also found in neuroendocrine tumors. Calcitonin secretion is known to be one of the main characteristics of MTC, a tumor originating from parafollicular neuroendocrine cells. Normal parafollicular cells commonly express calcitonin, but rarely express CGRP for which the process of alternative splicing is required[32]. However, a previous study observed a significant increase in CGRP mRNA in rapidly proliferating MTC tumor cells[52]. Consistent with previous findings, we confirmed high expression of CGRP in MTC by IHC[16]. Furthermore, MTC patients with high expression of CGRP had worse DFS, suggesting that CGRP may play an important role and may serve as a potential prognostic marker for MTC.

Because obtaining MTC tumors and their controls, including PTC tumors and adjacent normal thyroid tissues, is ethically feasible, MTC could serve as a good model to study the physiological function of CGRP in humans. MTC is an aggressive neuroendocrine tumor with poor prognosis and unsatisfactory response to traditional therapy. To date, the microenvironment of MTC remains largely unknown and has only been dissected in a few studies[53]. In the present study, we presented a comprehensive single-cell landscape of MTC and showed a lower immune infiltration in MTC when compared to PTC as well as some other common cancers and known "immune-cold" cancers, which is consistent with the decreasing number of lymphocytes in a previous IHC study of MTC[18]. It is well known that the immune microenvironment of a tumor can be characterized into inflamed (high immune infiltration), cold (low immune infiltration) and excluded (immune infiltrate residing only at the tumor margin) phenotypes based on the pattern of immune cell infiltration[54,55]. Comparison of distribution showed that all types of immune cells decreased in MTC tumor, but an increase was still observed compared to adjacent normal tissue. This suggests that MTC is an "immune-cold" tumor rather than an "immune-excluded" tumor that suppresses immune cell function and migration through peripheral stromal cells.

To investigate the effect of CGRP in immune modulation, we performed cell-cell communication analysis and multi-immunohistochemical staining, finding that tumor-produced CGRP affected DCs through the CGRP receptor CALCRL. Previous studies have shown that CGRP activates the cAMP pathway upon binding to CGRP receptors on DCs[34–36]. We found that a higher activation level of the cAMP pathway in DCs from MTC and further investigation on DCs by unsupervised re-clustering revealed a dysfunctional and tolerogenic state of DCs in MTC, characterized by reduced antigen-presenting and co-stimulatory functions as compared to PTC, which consistent with the result of low MHC-II expression DCs in previous mIHC study[19]. In tumors, DCs are mainly developed from infiltrated monocytes or DC progenitors in response to local cytokine networks[38]. Although previous animal and in vitro studies demonstrated that transient exposure of CGRP could suppress the maturation of well-developed DCs, the effect and the mechanism of CGRP on the development of DCs remains unclear[56]. Our pseudotime analysis suggested that the development of DCs in MTC was quite different from that in PTC, indicating that the CGRP may drive an abnormal development of DCs.

Further analysis using CellOracle revealed the changes of key transcriptional factor KLF2 during abnormal developmental trajectory of DCs. KLF2 is a transcriptional factor highly expressed in the resting/quiescent state of immune cells[40]. Identified as a repressor of myeloid cell activation, KLF2 deficiency could enhance host immune responses against polymicrobial infection[57]. Moreover, the inhibitory effect of KLF2 in DCs has been illustrated by knock-down experiments in vitro, where an increased capacity of KLF2-knock-down DCs was observed in proatherogenic immune responses[41]. Our pseudotime analysis suggested that KLF2 expression gradually decreased along with the normal developmental process of DCs as observed in PTC. In contrast, KLF2 expression level dropped at a much slower pace in MTC. These findings were subsequently validated by experiments. Furthermore, whether CGRP induces an increase in KLF2 levels and affects DCs function through the activation of the cAMP pathway remains unknown. Our experimental results demonstrated that KLF2 was quickly down-regulated during a differentiation process from monocytes to DCs, whereas the presence of CGRP slowed down the loss of KLF2 and finally disrupted the development of competent DCs. When we use the cAMP inhibitor SQ22536 to suppress downstream cAMP pathway activation, the expression of KLF2 influenced by CGRP decreased, and the suppressed DC function was restored.

According to our research findings, CGRP is a key factor in inducing changes in the immunosuppressive microenvironment of MTC. However, the precise mechanisms underlying its high expression remain unclear. Previous study reported that RET kinase inhibitors profoundly inhibit calcitonin levels of MTC patients and therefore likely also inhibit CGRP as well[58]. This inhibitory effect is not solely due to decreased tumor burden but interrupts a physiological regulatory role of RET on calcitonin gene expression[59]. Therefore, it is worth exploring whether the application of RET kinase inhibitors can effectively suppress CGRP levels and thereby reverse the immunosuppressive microenvironment of MTC. In that respect, sampling a tumor from a patient receiving selpercatinib would be an invaluable resource to demonstrate reversibility of the putative effects of CGRP on DCs in MTC.

On one hand, as previously mentioned, our research findings provide translational relevance for MTC, as it would support the rationale for combination trials of RET kinase inhibitors with immune checkpoint inhibitors for RET-mutant MTCs. On the other hand, of note, we also found that Rimegepant, a CGRP receptor antagonist, could block the effect of CGRP and restore a functional DC. Rimegepant is a American Food and Drug Administration (FDA) approved drug for preventing and treating the symptoms of migraine headaches. It has been considered well-tolerated and safe in clinical practice[60]. Further in vivo studies and clinical trials for the efficacy of Rimegepant combined with system therapy or other kinds of immunotherapy in MTC treatment are meriting. Besides MTC, CGRP has also been reported to be expressed in other types of neuroendocrine cancers such as small cell lung cancer[61,62], pheochromocytoma[63] and neuroendocrine prostate cancer[64–66]. Therefore, it would be interesting to know whether CGRP modulates the microenvironment of these tumors as well.

In conclusion, we have generated a comprehensive single-cell atlas of MTC, demonstrating the crucial role of neurotransmitter CGRP in shaping an immunosuppressive tumor microenvironment. Notably, CGRP drives an abnormal development of intratumoral DCs by activation of cAMP pathway and preventing the loss of the transcription factor KLF2. These results may facilitate the development of effective therapies for MTC, and provide human evidence of CGRP's function outside the nervous system.

## Methods

### Study approval

The study was approved by the Institutional Research Ethics Committee of The First Affiliated Hospital of Sun Yat-sen University ([2021] 109). Informed consent, including to publish clinical information potentially identifying individuals, was obtained from all patients.

### Clinical samples

Surgical samples of seven MTC and eight PTC for single-cell RNA sequencing were collected from the First Affiliated Hospital of Sun Yat-sen University. Clinical information of patients and the results of the statistical analysis were listed in Supplementary Data 5. Pathological paraffin section samples for IHC and mIHC were collected from the

First Affiliated Hospital of Sun Yat-sen University and Sun Yat-sen University Cancer Center including a 120 MTC and 61 age- and sex-matched PTC patients. All patients were pathologically confirmed by experienced pathologists.

## Single-cell RNA sequencing

**Cell preparation.** Fresh thyroid tissue samples were collected from surgery immediately and cut into 1-2 mm pieces after washing by PBS. Tissue samples in small pieces were then digested with Tumor Dissociation Kit (cat# 130-095-929, Miltenyi Biiotec) or Trypsin (cat# 25300062, ThermoFisher) for 15 min (37 °C). The cell suspension was collected after filtration through 70 μm MACS SmartStrainer (cat# 130-098-462, Miltenyi Biotec) and 30 μm MACS SmartStrainer (cat# 130-098-458, Miltenyi Biotec). RBC Lysis (cat#00430054, eBioscience™) was used to lyse red blood cells on ice after centrifugation at 400 x $g$ for 6 min. After washing with PBS for once, the collected cells were counted by AO/PI and the concentration was adjusted to 500–1300 cells/μL for library preparation. Peripheral blood mononuclear cells were isolated using Ficoll-Paque PREMIUM (cat# 17544203, Cytiva) at 1800 x rpm for 30 min centrifugation. We then collected the central medium of cells and lysed RBCs by RBC Lysis. To prepare the mixed cell suspension, we followed the manufacturer's instructions for "Chromium Next GEM Single Cell 5' Reagent Kits to capture cells per library" (10x Genomics, v2 chemistry). We added appropriate volume of nuclease-free water and the corresponding volume of single cell suspension for each sample tube.

**Preparation of single-cell RNA sequencing library.** After running for Gel Bead-In Emulsions (GEMs) generation and cell barcoding, GEMs were used for reverse transcription incubation, followed by cDNA amplification, quality control and quantification. Subsequently, 5' gene expression libraries and V(D)J libraries were constructed according to the manufacturer's standard in the 10x Genomics protocol (Single Cell 5' Reagent Kits v5.2 User Guide). All libraries were pooled and sequenced on Novaseq™ 6000 (Illumina, San Diego, CA).

**Data processing and clustering.** We used the Cellranger Single-Cell toolkit (v.6.1.1) for mapping reads to the human genome (GRCh38). The filtered feature barcode matrix was used for further data analysis. For quality control, several approaches were performed: (1) ambient cell free mRNA contamination was removed using SoupX v1.5.2 for each individual sample[67]; (2) low-quality cells with expressed genes < 200 or > 8000 were removed; (3) low-quality cells with mitochondrial genes more than 25% were removed. After removal of ribosomal and mitochondrial genes, we normalized and identified the top 1000–2000 highly variable genes by "vst" methods using the Seurat R package (v 4.0)[68]. To scale the expression of each gene, we applied "ScaleData" function and performed Principal Component Analysis (PCA) to determine an adequate number 16–30 for further "FindNeighbors" analysis. With the resolution set to 0.2-0.3, "FindCluster" and "RunUMAP" functions were used for dimensionality reduction and visualization. Doublets were recognized by simultaneous expression marker genes of two or more major cell types and removed from further analysis. We performed correction by R package Harmony (version 0.1.0) based on the corresponding top PCA components identified in the re-clustering subpopulation of non-malignant cells[69].

**Cluster annotation.** For major cell types and their subpopulations, we used "FindAllMarkers" function to find differentially expressed genes between each cluster by setting that genes expressed > 0.5 Log2 fold change threshold and detected in more than 25% in either of two populations. We then annotated the cell cluster according to the differentially expressed genes of each cluster. Cell type proportion data of public PTC[23,24], anaplastic thyroid cancer (ATC)[24,25], breast cancer (BC)[26], gastric cancer (GC)[27], hepatocellular carcinoma (HCC),

intrahepatic cholangiocarcinoma (ICC)[28], pancreatic cancer (PDAC)[29], glioblastoma (GBM)[30] and prostate cancer (PRAD)[31] were obtained from public data of published research.

**Infer-CNV analysis.** Specifically, tumor cells of MTC and PTC were annotated according to tissue originations and validated by calling chromosomal copy-number variations (CNVs), respectively. Using T cells as reference, CNVs analysis in single-cell data was performed by inferCNV (version 1.8.1, https://github.com/broadinstitute/inferCNV) R package with default parameters. According to the methods reported in previous study, a distribution map was constructed to verify the differences in correlation distribution of CNV signals between reference cells and tumor cells: compared with reference cells, tumor cells often appear in another peak[70].

**Comparison of cell distribution between groups.** To evaluate the distribution of cell types and their subpopulations between MTC and PTC, we calculated the ratio of observed cell numbers to expected cell numbers in each cluster using Chi-square test as previously reported[71]. $X^2$ was considered as chi-square value in the following equation, where $fe$ referred to expected cell numbers and $fo$ referred to observed cell numbers in a specific cell cluster. The ratio was calculated by observed cell numbers and expected cell numbers.

$$X^2 = \sum \frac{(fo - fe)^2}{fe} \qquad (1)$$

$$R_{o/e} = \frac{fo}{fe} \qquad (2)$$

**Differential expression analysis and pathway enrichment analysis.** Differential expression analysis of specific clusters or comparisons between MTC and PTC was performed by using the "FindAllMarker" function as described above and selected significant genes with Log-scaled fold change > 0.5 and $p$ value < 0.05. Gene set enrichment analysis was performed using the R package "clusterProfiler" (ver 4.0.5)[72] under Gene Oncology gene sets released by MSigDB[73–75]. Pathways with $p$ value < 0.05 were considered as significantly enriched.

**Gene signature score calculation.** Based on the published description of naive-like, cytotoxic and dysfunctional T cells[26,76,77], we used the "AddModuleScore" function in the Seurat package to calculate the naive-like score, cytotoxic score and dysfunctional score of CD8$^+$ T cells. Activation signatures combining cytotoxic and dysfunctional signatures were calculated in further correlation analysis. The gene signatures of antigen-presentation and co-stimulation of DCs were referenced from published works[78–80]. The gene signature of cAMP related pathways activation was referenced from cAMP or downstream PKA related pathways in GO dataset. In the tumor cell analysis, the thyroid differentiation score (TDS), which represents thyroid function, was taken from the paper published by TCGA on genomic researching of thyroid cancer[22]. The inhibitory gene signature of macrophage was referenced from previous studies[46,47]. All the gene signatures we used are shown in Supplementary Data 6.

**cNMF reclustering.** DCs were selected and re-clustered using a consensus non-negative matrix factorization (cNMF) and a graph cluster-based approach[81]. Then, 7 modules of DCs were identified. Reclustering of DCs from tumors and peripheral thyroid tissues according to tumor type was performed as above. Modules consisting of gene programs were defined and characterized by immune pathways (enriched by the top gene signature of each module). Since we aimed to discover MTC specific gene expression programs by cNMF, we decided to use cNMF-reclustering instead of PCA for further DCs analysis.

**Correlation analysis.** To confirm the relationship, we performed correlation analysis between the gene signatures of cNMF-derived DC modules and two functional scores of DCs, respectively. Further correlation analysis was performed between *KLF2* expression and co-stimulatory score, cAMP related pathways activation score of DCs. For the correlative analysis between the activation score of CD8⁺T cells and the co-stimulatory function of DCs, we used the same methodology as described above. Correlation analysis was performed by Pearson's test or Spearman's test.

**Cell-cell interaction analysis by cellchat.** To reveal possible cell-cell communication, we applied R package CellChat (v1.1.3) to detect significant ligand-receptor pairs in MTC and PTC, respectively. According to the protocol of the package, CellChat data were merged using the "mergeCellChat" function. We used "netVisual_bubble" function in displaying specific ligand-receptor interactions while "subsetCommunication" function was used to extract interaction scores of ligand-receptor pairs in both MTC and PTC.

**Pseudotime analysis by Monocle2.** Using Monocle2 R package[82], we inferred a developmental trajectory of CD8⁺T cells and DCs. We imported sequencing data into Monocle2 as CellDataSet class, we applied negative biominal distribution to count data and selected differentially expressed genes in cell populations. To reduce dimensionality, we performed reversed graph embedding algorithm and ordered cells in pseudo time with lower dimensional expression data. Trajectories of CD8⁺ T cells and DCs were constructed to show the developmental process of cells, respectively. Then, the differentially expressed genes along pseudotime development was analyzed by "differentialGeneTest" function and divided into two groups according to distributed location on trajectory.

**Key transcriptional factor analysis by CellOracle.** We build gene regulatory networks (GRNs) with CellOracle (v.0.12.0) following the tutorials on https://morris-lab.github.io/CellOracle.documentation/[39]. Single-cell data of monocytes and DCs was used in CellOracle analysis. We used the DDRTree dimensions analyzed by Monocle2 instead of recalculating cell developmental trajectory. The key transcription factors of each group were identified based on overlapped transcriptional factors of degree centrality, betweenness centrality and eigenvector centrality. The final selection of key transcription factors required to meet the following criteria: 1) Specific to State1 and without a significant role in State2; 2) Considered to be key transcription factors in MTC tumor. Simulating the effects of perturbing *KLF2* on gene expression in monocytes and DCs involved setting the expression of the *KLF2* to 0. CellOracle then utilized the simulated gene expression changes to predict the trajectory of cellular transition.

**TCR data processing and expansion analysis.** Single-cell RNA data were processed using the Single-Cell Toolkit vdj function (v.6.1.1, 10x Genomics Inc) to assemble VDJ receptor sequences. The filtered_contig_annotations.csv outputs from the samples were used for downstream analysis. We removed TCR α- and β-chain nucleotide sequences that did not meet the following criteria: (1) full-length; (2) with a valid cell barcode; (3) matched α/β chains. If more than one TCRα/β chains were detected in one cell, only the clonotype with the highest expression was retained. The median value of the cytotoxic score was chosen as a cut-off criterion to select cytotoxic T cells for further analysis. We identified cells with identical CDR3 amino acid sequence as clonal cells that derived from the same T cell clonotype.

**TCR cross tissue analysis.** Although we have already grouped cells according to CDR3 amino acid sequences in each sample, we also aimed to identify shared clonotypes across samples. After re-grouping clonotypes across the tumor, peripheral normal thyroid tissue and blood samples for each patient, clonotypes were assigned a tissue expansion pattern based on clone size in different tissue types as reported[43]. T cell clonotypes were called normal tissue singlet when having one cell in normal adjacent thyroid tissue but none in tumor, while clonotypes that having more than one cell in normal adjacent thyroid tissue but none in tumor were called normal tissue multiplets. Conversely, clonotypes with one cell in tumor but none in normal thyroid tissue were called Tumor singlet, while clonotypes with more than one cell in tumor but none in normal thyroid tissue were called tumor multiplets. Dual-expanded clones represented clonotypes that had at least one cell in both normal thyroid tissue and tumor. Clonotypes from blood samples were identified according to the same criteria as tumor and normal thyroid tissue, whether they had cells in normal tissue or tumor.

## Bulk-RNA sequencing

**Sample preparation and libraries and sequencing.** For bulk-RNA data of MTC , after isolation from fresh frozen tissue, total mRNA was cut into short fragments. To synthesis cDNA, we used random hexamer-primer in the first-strand cDNA synthesis, and buffer, dNTPs, RNase H and DNA polymerase I were used for the second-strand cDNA synthesis. The QIAQuick PCR Extraction Kit (Qiagen, Hilden, Germany) was used to purify short fragments, which were then solubilized in EB buffer for end reparation and poly (A) adjunction. After conjugation with sequencing adaptors, suitable fragments selected from cDNA were recognized as templates for PCR amplification. Finally, the library was sequenced on the HiSeq X TEN platform, generating 150 bp paired-end reads. By the way, the bulk-RNA data of PTC and pair normal tissue is selected from our previous study. In order to match age and sex with MTC patients, only 28 PTC patients with same sex and age ± 1 years were selected in further comparison analysis with MTC. We used the PTC bulk-RNA data generated by the TCGA Research Network: https://www.cancer.gov/tcga as additional validation dataset.

**Data processing.** Bulk-RNA fastq files were aligned to the human genome (hg38) after read quality qualification and sequencing adapter removal[83]. To measure gene expression abundance, the hisat2-RSEQC pipeline was used to evaluate fragments per kilobase of exon model per million mapped fragments (FPKM)[84]. To analyze alternative transcriptional splicing product of *CALCA*, we use salmon to align and recognized calcitonin and *CGRP* transcript.

**Immune infiltration and score calculation.** Immune infiltration and immune cell composition were predicted by ESTIMATE and CIBERSORT which characterize cell composition of complex tissues from gene expression[85]. To quantify the activity of the immune system, the cytotoxic, dysfunctional and antigen-presenting gene lists used in single-cell RNA sequencing[26,76,77]. Gene set variation analysis (GSVA) was applied to each sample[86]. A high score represented high expression of genes that corresponding to a more active immune response.

**Immunohistochemistry.** FFPE blocks were cut into 5 μm-thick sections, which were blocked with 10% normal goat serum for 30 min after deparaffinization. The following antibodies were used as primary antibodies: anti-CD45 antibody (CST13917, 1:500, Cell Signaling Technology), anti-calcitonin antibody (ab16697, 1:500, Abcam), anti-CGRP antibody (CST14959S, 1:1600, Cell Signaling Technology) as primary anti-bodies. The primary antibodies were incubated respectively overnight and then subsequently probed with secondary antibodies (DAKO DAB kit). All slides were scanned with KF-PRO Slide Scanner (Kfbio, China). For images processing and analysis, we used Qupath (v0.3.0) for initial processing and positive cell counting. Image J software was used to analyzed expression intensity of CGRP. We randomly selected 5 region of tumor and corrected the light density of each region we chose. Identified the stained-positive region by using

segmentation in HIS mode, we selected measurement-IOD/ area ($cm^2$) to calculate average of density (AOD) value. Keep the parameters unchanged and repeated all the steps for 5 regions and finally calculated the average of density of each sample.

**Multiplex immunofluorescence staining.** Slides of FFPE blocks were dewaxed in xylene, then rehydrated for 5 min in a graded series of 100%, 90%, 80% and 70% ethanol and fixed in 10% neutral buffered formalin. Six marker panel was composed of anti-CGRP antibody (CST14959S, 1:1600, Cell Signaling Technology), anti-CD11c antibody (ab52632, 1:200, Abcam), anti-CD8 antibody (CST70306S, 1:400, Cell Signaling Technology), anti-CD86 antibody (ab239075, 1:100, Abcam), anti-HLA-DR antibody (ab92511, 1:4000, Abcam) and anti-CALCRL antibody (MAB10044, 1:400, R&D). To visualize markers simultaneously, we stained the slides with PANO 7-plex IHC kit, cat 0004100100 (Panovue, Beijing, China). In the staining process, slides were boiled in pH 9.0 Tris-EDTA buffer (Solarbio, Beijing, China) by microwave for antigen retrieval. After blocking proteins for 10 min, primary antibodies were sequentially incubated for 30 min at 37 °C and then incubated with HRP-conjugated Ms + Rb secondary antibody and enhanced tyramide signal with Opal for fluorescence microscopy detection. Our marker panel was combined with fluorescent dyes as follows: anti-CGRP + Opal620, anti-HLADR + Opal520, anti-CD86 +Opal570, anti-CD8 +Opal540, anti-CALCRL +Opal650 and anti-CD11c +Opal690. After TSA operation, the slides were microwaved and stained with 4' −6'-diamidino-2-phenylindole (DAPI, SIGMA-ALDRICH) for 10 min.

**Multiplex immunofluorescence analysis.** TissueFAXS platform (TissueGnostics) were used in the acquisition of multispectral images set at 20 nm wavelength intervals from 420–720 nm with the same exposure time for fluorescence spectra acquisition. Unmixed by spectral libraries established from images of single-staining images, Multispectral images were process by the StrataQuest (TissueGnostics) software. Cells with a specific phenotype were identified and quantified using the TissueQuest software when detection cut-offs were set according to positive controls.

## In vitro experiment

**Monocyte isolation and DC induction.** PBMCs were isolated from whole-blood samples of healthy donors using Ficoll-Paque PREMIUM after density centrifugation. Monocytes were isolated from PBMCs using EasySep™ Human CD14 Positive Selection Kit II (STEMCELL, cat#17858) according to the manufacturer's protocol and cultured in RPMI 1640 medium ($0.5 \times 10^6$ cells/ mL) supplemented with 20 ng/ml IL-4 and 30 ng/ml GM-CSF. Monocytes were incubated for 6 days at 37 °C and 5% $CO_2$ condition and the culture medium was changed every other day. To obtain mature DCs, we used conventional DC maturation method including 24 h stimulation with cocktail (2000 IU/mL IL-6, 400 IU/mL IL-1β, 2000 IU/mL TNF-α, and 2 μg/mL PGE₂).

**The isolation and culture of CD8⁺T cell.** PBMCs were isolated according to the steps mentioned above. CD8⁺T cells were isolated from PBMCs using EasySep™ Human CD8⁺ T Cell Isolation Kit (STEMCELL, cat#17953) according to the manufacturer's protocol and cultured in RPMI 1640 medium ($0.5 \times 10^6$ cells/ mL) supplemented with 50 ng/mL IL-2 and Dynabeads™ Human T-Activator CD3/CD28 (Invitrogen). CD8⁺T cells were incubated for 4 days at 37 °C and 5% $CO_2$ condition and the culture medium was changed every other day. Pharmingen™ Leukocyte Activation Cocktail was added and cultured for 4 h to promote the expression of CD8⁺T cell functional molecules before flow cytometry.

**CGRP and Rimegepant treatment.** 400 nM/L CGRP was added from the beginning of DC induction. Cytokine cocktail was added to induce DC maturation. In some experiments, 200 nM/L Rimegepant was used against the CGRP receptor. Then, DCs were collected for further qPCR and flow cytometry analysis. We first reviewed relevant literature and found that the safe dose of Rimegepant is below 10 μM concentration, and 100 nM of Rimegepant can effectively restore the function influenced by CGRP[87]. Subsequently, we conducted preliminary experiments using concentrations of 100 nM, 200 nM, and 400 nM. Based on the optimal recovery results, we selected a concentration of 200 nM Rimegepant.

**ELISA assay of cAMP concentration in CGRP treated DCs.** On the sixth day of DC induction, 25 μM Rolipram was added 30 min to control group and CGRP group to inhibit phosphodiesterase and prevent the degradation of intracellular cAMP before treatment. 400 nM CGRP was added to the treatment group and cultured for 30 min. Subsequently, DCs were lysed and ELISA experiments were performed according to the official instructions of the cAMP Assay Kit (Competitive ELISA, Fluorometric, cat#ab138880). Finally, we monitored fluorescence increase at Ex/Em = 540/590 nm (cutoff 570 nm) using a Thermo Scientific Varioskan Lux microplate reader in top read mode.

**Isolation of DC RNA, synthesis of cDNA, qPCR analysis.** Trizol reagent (Invitrogen) was used to isolate total RNA from DCs from in vitro experiments. RNA was then precipitated in isopropanol. cDNA was synthesized using PrimeScript™ RT Master Mix (Takara). The specific primer was synthesized as follows: *β-actin* (forward =5′-CATGTACGTTGCTATCCAGGC-3′, reverse =5′-CTCCTTAATGTCACGC ACGAT-3′); *KLF2* (forward=5′-CTACACCAAGAGTTCGCATCTG-3′, reverse=5′-CCGTGTGCTTTCGGTAGTG-3′). PCR was performed in triplicate using Taq Pro Universal SYBR qPCR Master Mix (Vazyme) in the LightCycler 480 real time PCR system (Roche). All results are expressed in arbitrary units relative to *β-actin* RNA expression.

**CD8⁺T cell proliferation assay.** CD8⁺T cells were isolated as mentioned above and then labelled with carboxyfluorescein succinimidyl ester (CFSE; Invitrogen) in the presence of recombinant IL-2 and CD3/CD28 dynabeads for 4 days. CD8⁺T cells with or without CD3/CD28 dynabeads was used as positive or negative control. CFSE signal was acquired by flow cytometry using FACS Fortessa X-20 (BD Biosciences).

**Flow cytometry.** Flow cytometry analysis was performed using CYTEK Aurora flow cytometry and Flow Jo v10.6.2 software (Tree Star) was used for data analysis. DCs were stained in PBS containing Zombie Live-Dead dye or Live-Dead dye (FVS-780) for 20 min and then in FACS buffer with antibody cocktails including CD45 (cat#563792, BUV395, BD), CD11c (cat#561356, PE-Cy7, BD), CD86 (cat#566131, BV480, BD), CD83 (cat#305336, BUV605, BD), CD80 (cat#305216, AF647, BD), CD40 (cat#334305, FITC, BD), HLA-DR (cat#748338, BUV805, BD) on ice for 20 min. CD8⁺T cells were stained in PBS containing Live-Dead dye (FVS-780), CD3 (cat#612895, BUV805, BD), CD8 (cat# 563823, BV786, BD Pharmingen) for 20 min and then stained for IFN-γ (cat# 554700, FITC, BD) and GZMB (cat# 562462, PE-CF594, BD) for an hour. The staining reaction was terminated by the addition of 500 ul PBS or FACS buffer. Stained cells were fixed with 1% paraformaldehyde and then analyzed on the CYTEK Aurora. The detailed information was provided as key resource table in Supplementary Data 8.

**Survival analysis.** 102 MTC patients with prognostic information were divided into high and low groups according to the median value of CGRP expression intensity and their clinical information was provided in Supplementary Data 7. Disease-free survival was analyzed with a two-sided log-rank test, with the hazard ratio (HR) and two-sided 95%

CIs based on a Cox proportional-hazards model and the associated Kaplan-Meier survival estimates using R package Survival (v3.2.11) and Survminer (v0.4.9).

**Statistical analysis.** All data analyses were performed in R 4.0.2 and statistical significance was defined as a two-tailed $P$ value of less than 0.05 by Wilcoxon test or $t$-test as description in comparison of two groups. ANNOVAR was used to compare more than two groups in experimental data.

## Reporting summary

Further information on research design is available in the Nature Portfolio Reporting Summary linked to this article.

## Data availability

The transcriptomic data reported in this study have been deposited at the Genome Sequence Archive at the National Genomics Data Center (Beijing, China) under the Accession ID HRA006084. Dataset HRA006084 is available under restricted access because of data privacy and supervision. For research purpose, access can be obtained by the DAC (Data Access Committees) of the GSA-human database. The approximate response time for accession requests is about one month. Once access has been approved, the data will be available for two months. The single-cell publicly available data used in this study are available in the GEO database under accession code GSE163558, GSE176078, GSE151530, GSE148673, GSE193581, GSE191288, GSE148673, GSE217845, GSE223063, GSE181294. The TCGA public transcriptome data used in our study are available in thyroid cancer part in UCSC xena dataset [https://xenabrowser.net/datapages/]. The remaining data are available within the Article, Supplementary Information or Source Data file. Source data are provided with this paper.

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

## Acknowledgements

We thank Dongming Kuang, Fang Wang, Shuling Chen and Zhonghui Tang for suggestions to our study. We thank Mengmeng Zhang, Bei Jin, Cheng Cheng, Huanjing Hu and Zhihang Chen for help with bioinformatical analysis. We thank Xiaoxue Ren, Xiaofei Liu, Guopei Zhang, Changyi Liao and Xiaofeng Huang for assistances in sample collection and in vitro experiments. This study was supported by the grant from National Natural Science Foundation of China (82271776, 82103035 and 82203461) and Guangzhou Science and Technology Project (2022342). Figure 1A and Fig. 7C were created with Biorender.com.

## Author contributions

Conceptualization: Y.H.L., H.P.X. Sample collection: Y.T.H., B.L., W.N.C., W.M.L. Methodology: Y.T.H., J.J., T.Y.X., X.W.C., X.Y.L., J.Y.Z. Investigation: Y.T.H., S.L.L., G.R.L., Y.Q.W. Visualization: T.Y.X., X.X., Y.B.X., J.P.W. Supervision: S.P., Y.H.L. Writing—original draft: Y.T.H.,Y.H.L. Writing—review & editing: H.P.X., J.W., Y.H.L.

## Competing interests

The authors declare no competing interests.
