## [Peer Review File · Nature Communications]

The neurotransmitter calcitonin gene-related peptide shapes an immunosuppressive microenvironment in medullary thyroid cancerREVIEWER COMMENTS

Reviewer #1 (Remarks to the Author): with expertise in thyroid cancers

Hou et al compared the immune landscape of MTC & PTC tumors by single cell sequencing and identify a role of the MTC tumor cell-derived calcitonin isoform CGRP in modulating the immune TME. Notable findings: 1. Compared to other thyroid cancer types MTCs present an immunologically cold tumor phenotype. 2. MTC tumor cells express CGRP that interacts with DCs through its receptor CALCRL, which is associated with de-repression of KLF2, a TF that interferes with DC maturation impairing antigen presentation and T-cell activation. 3. In vitro evidence shows that CGRP impairs monocyte differentiation to immature DCs, associated with attenuated decline in expression of KLF2. Overall, this is a carefully conducted study. The computational analyses of the single cell and bulk transcriptomic data is exhaustive and supports the conclusions. The attempt to provide orthogonal evidence through in vitro experiments goes part way to support some of these findings.

1. The most relevant prior study aligning with one of the key conclusions of this work is the paper in J Immunology, quoted by the authors, showing that KLF2 KO in mouse DC promotes DC maturation, antigen presentation and inflammation in the atherosclerotic plaque. The relationship between CGRP, cAMP signaling and KLF2 is more novel, and could be explored in greater detail to provide greater confidence in the key findings of this study. Specifically, is there any evidence that cAMP signaling prevents or delays the DC maturation process, and that it operates via regulation of KLF2?

2. Along the same lines, delayed maturation would also result in decreased migration of DCs to LNs. Were lymph nodes sampled and studied?

3. Were there sufficient cells available to discriminate whether CALCRL is preferentially expressed in cDC1 vs cDC2 cells?

4. The authors seem to entirely dismiss the MTC TME paper by Pozdeyev et al (Thyroid 2020). They refer to the work (ref 17), but do not acknowledge its findings. The analysis by Pozdeyev is through multispectral IHC. They found that DCs are MHCII low, consistent with the findings in this paper. Moreover, they found major differences in cDC abundance between MTC tumors (see Fig 4). Please indicate between-sample variability in DC cells sequenced.

5. Information on the mutation status, tumor mutation burden and TNM stage of the

cancers included in the study would be worth including.

6. In Fig. 1B, it would be informative if authors identify key functional myeloid subtypes (e.g. M1- and M2-like macrophages, PDL1+ or other dysfunctional markers) and demonstrate proportions.

7. Is the calcitonin alternatively spliced isoform CGRP universally expressed in all MTCs? Fig. 2E shows differential mortality based on CGRP expression. Does CGRP expression correlate with genotype (e.g. RET vs RAS, other genotypes?).

8. Fig. 2F: Please show the complete cell chat interactions in the supplementary. The authors should consider showing key phenotypes like CALCA-CALCRL interaction by CellChat and CGRP expression by IHC in all 7 MTCs (sample-by-sample in the Supplementary Figure) to demonstrate that the key phenotypes are not skewed by a few samples in the study cohort.

9. In the introduction the authors indicate correctly that MTCs do not respond to chemotherapy but paint an unwarranted bleak picture of their overall response to therapy, despite important recent therapeutic advances. Although the efficacy of multikinase inhibitors and selective RET kinase inhibitors has so far been demonstrated based on overall responses and progression free survival, early indications are that these drugs do also prolong overall survival (Schlumberger M et al. *Ann Oncol.* 2017 Nov; 28(11): 2813–2819).

10. In that regard, RET kinase inhibitors profoundly inhibit calcitonin levels (Wirth L et al. *NEJM* 2020) and therefore likely also inhibit CGRP as well. This inhibitory effect is not solely due to decreased tumor burden but interrupts a physiological regulatory role of RET on calcitonin gene expression (Akeno-Stuart N et al. *Cancer Res.* 2007). Hence, advocating CGRP receptor antagonists as a therapeutic strategy is not likely to gain traction in the field, since profound inhibition of the ligand via RET kinase inhibition is already an approved therapy. In that respect, sampling a tumor from a patient receiving selipercatinib would be an invaluable resource to demonstrate reversibility of the putative effects of CGRP on DC and T cells. This is feasible, since neoadjuvant clinical trials of selipercatinib are currently accruing, so the surgical samples will be available for study. This has translational relevance, as it would support the rationale for combination trials of RET kinase inhibitors with immune checkpoint inhibitors for RET-mutant MTCs. This information could be added to the discussion.

Reviewer #2 (Remarks to the Author): with expertise in bioinformatics, thyroid cancers

Hou et al. reported the roles of Neurotransmitter calcitonin gene-1 related peptide in MTC. However, throughout the manuscript, the authors mostly analyzed the difference of two genes in MTC compared to PTC. Obviously, hundreds of genes could be differentially regulated in the two cancer subtypes. Since the authors' manuscript aimed to report the roles of two specific genes, particularly their encoded peptide, more functional assays need to be pursued.

First of all, the authors used gene names and protein names interchangeably, which caused a lot of confusions.

Next, the manuscript is titled: "Neurotransmitter calcitonin gene-1 related peptide shaped an immunosuppressive microenvironment in medullary thyroid cancer". However, there was not any functional test of Neurotransmitter calcitonin gene-1 related peptide throughout this manuscript. Indeed, the coding gene of this peptide is among the long list of genes that are altered in MTC compared to PTC and is associated with tumor-immune interactions, however, the data in this manuscript couldn't support the functions of the peptide encoded by this gene.

What is the evidence of tumor and normal cell classifications? InferCNV results are blurry and color keys are not one-to-one mapped. The reference seemed to have tumor cells. Are MTC expected to have strong aneuploidy?

Lines 135-154, what were statistics? Were the immune "cold" observations due to sampling bias? What are the relative immune cell proportions compared to other well-known immune "cold" tumor types?

The authors analyzed 8 MTC and 7 PTC tumors, however, there are multiple thyroid cancer datasets published, including scRNAseq of PTC and ATC, and bulk RNAseq of TCGA studies. What are the comparisons between the newly generated data to previously published larger cohorts?

Reviewer #3 (Remarks to the Author): with expertise in thyroid cancers, immunology

The authors used single-cell RNA sequencing to analyze the composition and functional status of tumor and immune cells in PTC and MTC, revealing the role of CGRP on DC development and immunosuppressive landscape in MTC. The possible connection between CGRP/DC/KLF2 and T cell is intriguing. This manuscript could be improved by addressing the following comments:

1) A key question is whether there is a direct interaction between CGRP and tumor-infiltrating T cells. It would benefit from cell experiments with CGRP and T cells.

2) How to determine the concentration of Rimegepant against the CGRP receptor?

3) In Fig 1G, the samples used for IHC confirmation were not specified in the methods section. It would be better to provide the patients' characteristics of these samples.

4) In Fig 2E, were the same samples used in Fig 1G? How to calculate the score of CGRP expression?

5) Fig 4I did not show similar trends between MTC expressing CGRP and PTC without CGRP expression. This should be clarified.

Response to Reviewers' comments

Neurotransmitter calcitonin gene-related peptide shaped an immunosuppressive microenvironment in medullary thyroid cancer

We sincerely appreciate that all three reviewers have provided constructive comments and helpful advices for manuscript refinement. Following the reviewers' kind instructions, we have modified our inadequate descriptions and supplied new data to support our conclusion and address the reviewers' concerns. We provided more evidences to validate the involvement of CGRP in shaping the immunosuppressive TME and enriched our understanding towards immune cell interaction mechanism by additional experiments. Published cohorts of single-cell RNA sequencing and TCGA dataset were also added in our analysis and clinical characterization of MTC patients' samples were complemented.

Please find our point-by-point responses to the reviewers' comments as followed:

Reviewer #1: Major Comments:

- The most relevant prior study aligning with one of the key conclusions of this work is the paper in *J Immunology*, quoted by the authors, showing that KLF2 KO in mouse DC promotes DC maturation, antigen presentation and inflammation in the atherosclerotic plaque. The relationship between CGRP, cAMP signaling and KLF2 is more novel, and could be explored in greater detail to provide greater confidence in the key findings of this study. Specifically, is there any evidence that cAMP signaling prevents or delays the DC maturation process, and that it operates via regulation of KLF2?

Our response: We appreciate your valuable suggestions. To provide further evidence for the relationship between CGRP, cAMP signaling, and KLF2, we first conducted analyses based on our dataset integrated with public single-cell datasets of thyroid cancer. As shown in Supplementary Figure 2F, the activation scores of the cAMP pathway in dendritic cells (DCs) from medullary thyroid carcinoma (MTC) tumors were higher compared to those from papillary thyroid carcinoma (PTC), public datasets of PTC and anaplastic thyroid carcinoma (ATC). When analyzing all thyroid cancer tumor samples collectively, we also found a positive correlation between the activation score of the cAMP pathway and the expression level of *KLF2* in DCs, while showing a negative correlation with the co-stimulatory function of DCs (Supplementary Figure 5A-C).

Supplementary Figure 2F. Violin plot showing the genes signature score of cAMP related pathways activation in DCs derived from tumor of MTC, PTC and public dataset of PTC and ATC. Calculated by Wilcoxon test, * $p < 0.05$, ** $p < 0.01$, *** $p < 0.001$.

Supplementary Figure 5A-C. (A) Scatter plot showing the correlation relationship between KLF2 expression and cAMP activation level of tumor DCs in our cohort. (B) Scatter plot showing the correlation relationship between KLF2 expression and cAMP activation level of tumor DCs when integrated with public single-cell dataset. (C) Scatter plot showing the correlation relationship between DC co-stimulatory score and cAMP activation level of tumor DCs when integrated with public single-cell dataset.

Previous studies have demonstrated that upon binding to its receptor, CGRP activates the cAMP pathway through G-protein-coupled receptor signaling, leading to an increase in intracellular cAMP levels (Walker, C., 2010, Trends Pharmacol Sci). Adenylate cyclase is the primary enzyme of generating cAMP. Consistent with these findings, our *in vitro* cell experiments using ELISA showed that treatment with CGRP resulted in elevated levels of cAMP in DCs, as depicted in Supplementary Figure 2G.

Supplementary Figure 2G. ELISA assay showed the concentration of cAMP (nM) after CGRP treatment, ** $p < 0.01$.

SQ22536 is an inhibitor of adenylate cyclase in the cAMP pathway, effectively blocking the production of intracellular cAMP. To further elucidate the impact of CGRP on *KLF2* expression through the cAMP pathway and its inhibition of DC function, we added a CGRP+ SQ22536 group in our long-term CGRP-treated DC experiments. As depicted in Figure 5C, on the sixth day of DC induction, the *KLF2* levels in the CGRP long-term treatment group were significantly elevated, whereas the use of the CGRP receptor inhibitor group and SQ22536 inhibitor group reduced the elevated *KLF2* levels induced by CGRP. Additionally, flow cytometry experiments were conducted to analyze the expression of functional molecules in mature DCs. As shown in Figure 5D and Supplementary Figure 5F, the addition of SQ22536 effectively reversed the inhibitory effect of long-term CGRP treatment on DC function.

Figure 5C. Real-time PCR analysis of *KLF2* expression at day 6 during DC in different groups, ** $p < 0.01$, **** $p < 0.0001$.

Figure 5D. Representative flow cytometry histogram and increasing degrees of mean fluorescence intensity (MFI) of co-stimulatory markers CD80, CD40, CD86 and HLA-DR expressed on DCs. p values between groups were calculated by ANNOVAR, * $p < 0.05$, ** $p < 0.01$, *** $p < 0.001$, **** $p < 0.0001$.

Supplementary Figure 5F. Representative flow cytometry histogram and mean fluorescence intensity (MFI) of CD83 expression on mature DCs. p value was calculated by one-way ANNOVAR compared to control group, *** $p < 0.001$, **** $p < 0.0001$.

- Along the same lines, delayed maturation would also result in decreased migration of DCs to LNs. Were lymph nodes sampled and studied?

Our response: We appreciate the reviewer's suggestion. We also consider it very intriguing to explore whether the maturation impairment of DCs affects their ability to migrate to surrounding lymph nodes. But unfortunately, in clinical practices, lymph nodes are often small in size. In consideration of the primary requirement for pathological diagnosis, we did not obtain enough lymph node samples for single-cell sequencing.

- Were there sufficient cells available to discriminate whether *CALCRL* is preferentially expressed in cDC1 vs cDC2 cells?

Our response: Thank you for your question. *CALCRL* is preferentially expressed in cDC1. Based on single-cell data, we analyzed the expression proportions of CGRP receptor *CALCRL* and co-receptor *RAMP1* in different DC subsets. As shown in the figure below, receptors are expressed in all DC subsets. Specifically, the expression proportion in cDC1 is approximately 51%, in cDC2 it is around 16%, and in LAMP3⁺DC it is about 25%.

Left: UMAP plot show the expression of CGRP receptor among different subtypes of DC. Right: The barplot showing the proportion of receptor expressed DCs among different subtypes of DC.

- The authors seem to entirely dismiss the MTC TME paper by Pozdeyev et al (Thyroid 2020). They refer to the work (ref 17), but do not acknowledge its findings. The analysis by Pozdeyev is through multispectral IHC. They found that DCs are MHCII low, consistent with the findings in this paper. Moreover, they found major differences in cDC abundance between MTC tumors (see Fig 4). Please indicate between-sample variability in DC cells sequenced.

Our response: Thank you for your question. We have added relevant studies on the MTC immune microenvironment and revised descriptions that could potentially cause misunderstanding. “Currently, research on the immune microenvironment of MTC is relatively limited, primarily focused on studies based on immunohistochemistry (IHC) or multiplex immunohistochemistry (mIHC) staining results, without consensus on the conclusions. A study in 2017 suggested low PD-1-positive staining and low immune infiltration in MTC (Bongiovanni, M., 2017, *Endocr Relat Cancer*). However, a study published by Pozdeyev in 2020 using mIHC showed that 49% of primary lesions in MTC exhibited immune infiltration (mostly scattered or clustered around the tumor, with 14.6% cases observed within the tumor), which considered MTC was more immunologically than previous report (Pozdeyev, N., 2020, *Thyroid*).” (Page 6-7, Line 97-105).

As you mentioned, our results partially validated the findings from the study by Pozdeyev, which reported a reduced MHCII expression in DCs. “...revealed a dysfunctional and tolerogenic state of DCs in MTC, characterized by reduced antigen-presenting and co-stimulatory functions as compared to PTC, which consistent with the result of low MHC-II expression DCs in previous mIHC study.” (Page 26-27, Line 543-546)

The Supplementary Figure 7K illustrated a significant lower DC proportions in MTC compared with PTC tumor samples at the sample level, showing a relatively small variation in DC proportions among MTC tumor samples (Mean \pm Standard Deviation in MTC: 0.56% \pm 0.42%; in PTC: 1.67% \pm 1.1%).

*Supplementary Figure 7K. Box plot presented the proportion of DCs, *p<0.05.*

- Information on the mutation status, tumor mutation burden and TNM stage of the cancers included in the study would be worth including.

Our response: Thank you for your suggestion. We further performed whole-exome sequencing on the patients who underwent single-cell sequencing. The information of gene mutations and TMB burden have been added to Supplementary Table 5.

Patient ID	Mutation Status	Germline/Somatic	TMB
MTC01	RET M918T	somatic	23.3
MTC02	RET G691S	germline	3.5
MTC03	RET C634Y	germline	8.02
MTC04	RET C634R	germline	0.92
MTC05	HRAS Q61K	somatic	4.88
MTC06	RET C634Y	germline	1.4
MTC07	-	-	0.92
PTC01	-	-	1.24
PTC02	BRAF V600E	germline	1.68, 0.98
PTC03	BRAF V600E	somatic	1.5
PTC04	-	-	1.98, 1.94
PTC05	-	-	2.46
PTC06	-	-	0.94
PTC07	BRAF V600E	somatic	2.2
PTC08	BRAF V600E	somatic	1.9

Mutations and TMB information of patients for single-cell RNA sequencing from Supplementary Table5. NA indicates no known pathogenic mutations detected. The two numbers in TMB represent the values for the two tumor lesions from the same patient.

- In Fig. 1B, it would be informative if authors identify key functional myeloid subtypes (e.g. M1- and M2-like macrophages, PDL1+ or other dysfunctional markers) and demonstrate proportions.

Our response: Thank you for your suggestion. We have supplemented analyses on macrophages and incorporated it into the main text. “Macrophages also constitute an important population of myeloid cells with antigen presentation and immune regulatory roles other than DCs. Further analyses were performed and characteristic analysis based on subtypes showed that IL-1B⁺Macrophages had the highest M1 (pro-inflammatory subtype) score, while CCL18⁺Macrophages had the highest M2 (anti-inflammatory subtype) characteristics scores (Supplementary Figure 7C-D). Referring to previously published studies on the expression of immunosuppressive gene signature in macrophages, the results suggested that CCL18⁺Macrophages

might be a subtype with certain immunosuppressive functions, as their scores for immunosuppression-related genes were the highest among the three subtypes (Supplementary Figure 7E). Further analysis of the distribution differences of the three macrophage subtypes between MTC and PTC revealed that all three subtypes of macrophages were relatively more abundant in PTC tumors (Supplementary Figure 7F). Scoring analysis of macrophages based on the tissue source of MTC and PTC showed that both M1 and M2 scores of macrophages in MTC were slightly lower than those in PTC (Supplementary Figure 7G-H). Comparative analysis showed no significant difference in immunosuppression-related gene score and genes was observed in MTC macrophages (Supplementary Figure 7I-J). The lack of differences in macrophages between the two types of tumors suggested that macrophages may not be the primary cell population responsible for significant immunosuppressive responses in MTC.” (Page 22-23, Line 447-468)

Supplementary Figure 7C-J. (C-E) Violin plot showing the genes signature score of M1, M2 and immune inhibition in macrophage subtypes, *** $p < 0.001$. (F) Heatmap of the distribution ratio of myeloid cell subtypes in MTC and PTC tumor, * $p < 0.01$. (G-I) Violin plot showing the genes signature score of M1, M2 and immune inhibition in tumor macrophages derived from MTC and PTC, * $p < 0.01$, *** $p < 0.001$. (J) Violin plot showing the gene expression of immune inhibition in tumor macrophages derived from MTC and PTC.

- Is the calcitonin alternatively spliced isoform CGRP universally expressed in all MTCs? Fig. 2E shows differential mortality based on CGRP expression. Does CGRP expression correlate with genotype (e.g. RET vs RAS, other genotypes?).

Our response: Thank you for your suggestion. In our dataset, obvious expression of CGRP (the alternative splicing product of calcitonin) was observed in the tumor regions of 91 out of 102 patients with MTC. We also aimed to explore the correlation between CGRP expression and patient mutation genotypes. However, it is regrettable that genetic mutation information was not available for all patients. In our country, patients can autonomously decide whether to undergo molecular testing based on personal preferences, medical advice, economic conditions, and other factors. Therefore, in our study, we were unable to obtain molecular testing data for all MTC patients. Based on the existing genetic mutation information from single-cell sequencing of patients (5/7 cases with *RET* mutation, 1/7 cases with *HRAS* mutation and 1/7 case with no relevant pathogenic mutation), and considering that majority of patients have *RET* gene mutations, we have not yet observed a correlation between CGRP expression and genotype.

- Fig. 2F: Please show the complete cell chat interactions in the supplementary. The authors should consider showing key phenotypes like CALCA-CALCRL interaction by CellChat and CGRP expression by IHC in all 7 MTCs (sample-by-sample in the Supplementary Figure) to demonstrate that the key phenotypes are not skewed by a few samples in the study cohort.

Our response: Thank you for your suggestion. We have provided comprehensive CellChat interaction networks in the Supplementary Figure 2D, along with an additional analysis showing significantly different receptor-ligand interactions in MTC compared to PTC (Supplementary Figure 2E).

D

E

Supplementary Figure 2D-E. (D) Complete cellchat interaction plot between tumor cell and DCs was shown. (E) A dot plot was used to illustrate the upregulated receptor-ligand pairs in MTC compared to PTC, with the fold change on the right indicating the ratio of differences.

Furthermore, we have included IHC staining images of CGRP in the tumor regions of seven MTC patients used for single-cell sequencing in Supplementary Figure 2C.

Supplementary Figure 2C. In the tumor region of MTC patients for single-cell sequencing, IHC staining for CGRP was shown.

- In the introduction the authors indicate correctly that MTCs do not respond to chemotherapy but paint an unwarranted bleak picture of their overall response to therapy, despite important recent therapeutic advances. Although the efficacy of multikinase inhibitors and selective RET kinase inhibitors has so far been demonstrated based on overall responses and progression free survival, early indications are that these drugs do also prolong overall survival.

Our response: Thank you for your suggestion. We have revised the description of targeted therapy for MTC in the introduction section and included the clinical research results of the third-generation targeted drug selpercatinib for MTC treatment. “Multi-target tyrosine kinase inhibitors (TKIs) vandetanib and cabozantinib had been suggested for advanced MTC by American Thyroid Association. Although there is study found that TKIs could effectively improve patients’ overall survival (Schlumberger M et al., 2017, *Ann Oncol*), recent research results indicated that both vandetanib and cabozantinib, as well as the new generation of highly selective TKI selpercatinib (Hadoux, J. et al., 2023, *N Engl J Med*), can effectively enhance patients’ progression-free survival, whether overall survival can be improved remains to be observed (Elisei, R. et al., 2013, *J Clin Oncol*; Kreissl, M. C. et al., 2020, *J Clin Oncol*; Wells, S. A., Jr. et al., 2012, *J Clin Oncol*). However, TKI treatment is accompanied by a significant high incidence rate of adverse reactions (38.9-72%), along with the common issue of resistance to long-term treatment, necessitating extensive follow-up and deeper mechanistic studies.” (Page 6, Line 80-90)

- In that regard, RET kinase inhibitors profoundly inhibit calcitonin levels (Wirth L et al. NEJM 2020) and therefore likely also inhibit CGRP as well. This inhibitory effect is not solely due to decreased tumor burden but interrupts a physiological regulatory role of RET on calcitonin gene expression (Akeno-Stuart N et al. Cancer Res. 2007). Hence, advocating CGRP receptor antagonists as a therapeutic strategy is not likely to gain traction in the field, since profound inhibition of the ligand via RET kinase inhibition is already an approved therapy. In that respect, sampling a tumor from a patient receiving selpercatinib would be an invaluable resource to demonstrate reversibility of the putative effects of CGRP on DC and T cells. This is feasible, since neoadjuvant clinical trials of selpercatinib are currently accruing, so the surgical samples will be available for study. This has translational relevance, as it would support the rationale for combination trials of RET kinase inhibitors with immune checkpoint inhibitors for RET-mutant MTCs. This information could be added to the discussion.

Our response: Thank you for your valuable suggestion. We have added this section to the discussion in the main text. “According to our research findings, CGRP is a key factor in inducing changes in the immunosuppressive microenvironment of MTC. However, the precise mechanisms underlying its high expression remain unclear. Previous study reported that *RET* kinase inhibitors profoundly inhibit calcitonin levels of MTC patients (Wirth L et al. NEJM. 2020) and therefore likely also inhibit CGRP as well. This inhibitory effect is not solely due to decreased tumor burden but interrupts a physiological regulatory role of *RET* on calcitonin gene expression (Akeno-Stuart N et al. Cancer Res. 2007). Therefore, it is worth exploring whether the application of RET kinase inhibitors can effectively suppress CGRP expression and thereby reverse the immunosuppressive microenvironment of MTC. In that respect, sampling a tumor from a patient receiving selpercatinib would be an invaluable resource to demonstrate reversibility of the putative effects of CGRP on DCs in MTC.” (Page 28, Line 575-585)

Reviewer #2: Major Comments:

- First of all, the authors used gene names and protein names interchangeably, which caused a lot of confusions.

Our response: Thank you for your suggestion. We have made the modifications, where genes will be presented in italics and proteins in regular font in the main text.

- Next, the manuscript is titled: “Neurotransmitter calcitonin gene-1 related peptide shaped an immunosuppressive microenvironment in medullary thyroid cancer”. However, there was not any functional test of Neurotransmitter calcitonin gene-1 related peptide throughout this manuscript. Indeed, the coding gene of this peptide is among the long list of genes that are altered in MTC compared to PTC and is associated with tumor-immune interactions, however, the data in this manuscript couldn’t support the functions of the peptide encoded by this gene.

Our response: Thank you very much for your constructive question. Due to the clinical demand for advancements in treatment of MTC, it is significant to elucidate the MTC immune microenvironment characteristics and their underlying mechanisms. Because of lacking available animal models for functional studies in MTC, our study performed a comparative analysis of clinical samples from MTC and PTC, combined with *in vitro* functional validation methods. Based on the immunosuppressive microenvironment features by inactivated status of CD8⁺ T cells in MTC, we first conducted correlation analysis and found a positive correlation between the activation level of CD8⁺ T cells with both proportion and function of DCs in tumors (Supplementary Figure 7K-M). This suggested that DC dysfunction may be the key to the immunosuppressive microenvironment especially for CD8⁺T cell inhibition.

Supplementary Figure 7K-M. (K) Box plot presented the proportions of sub-populations of myeloid cells, $*p < 0.05$. (L) Scatter plot showed the correlation relationship between the proportion of myeloid sub-populations and the activation score of CD8⁺T cell at sample level. r means the correlation coefficient calculated by Pearson's correlation test. (M) Lollipop plot showed the correlation relationship between co-stimulatory score or antigen-presenting score with the activation score of CD8⁺T cell at sample level. r means the correlation coefficient calculated by Pearson's correlation test, $*p < 0.05$.

Furthermore, we analyzed the key cell populations that may affect DC development and function, by calculating a negative regulation of DC differentiation score in all cell types and correlating it with DC co-stimulatory function (Supplementary Figure 3J-K). Correlation analysis indicated that only the negative regulation score in tumor cell showed a significant negative correlation with DC co-stimulatory function (Supplementary Figure 3M). Further comparison revealed that the negative regulatory score of MTC tumor cells was significantly higher than that of PTC tumor cells (Supplementary Figure 3L). This indicated that MTC tumor cells are likely to be important cellular components affecting DC differentiation. To explore the influence of MTC tumor cells on DCs, we conducted interaction analysis between the two. We performed differential receptor-ligand pair screening from all receptor-ligand pair results. Among all interactions, compared to PTC, the CALCA_CALCRL interaction pair was the most obviously upregulated interaction between MTC

tumors and DCs (Supplementary Figure 2D-E). We further supplemented the results with analysis based on whole transcriptome data to demonstrate the relationship between *CGRP* and the immune microenvironment in MTC. The correlation analysis revealed that the expression level of *CGRP* in tumors was negatively correlated with the immune infiltration score, the cytotoxicity score of CD8⁺ T cells, the dysfunction score of CD8⁺ T cells, and the co-stimulatory function score of DCs (Figure 7B). The above results demonstrated at the genomic level how *CGRP* was identified and *CGRP* may be crucial in the dysfunction of DCs leading to the immune suppressive microenvironment in MTC.

Supplementary Figure 3J-M. (J) Violin plot showed negative regulation of DC differentiation score in all cell types in PTC tumor. (K) Violin plot showed negative regulation of DC score in all cell types in MTC tumor. (L) Negative regulation of DC score in MTC tumor cells and PTC tumor cells were displayed by violin plot. Calculated by Wilcoxon test, *** $p < 0.001$. (M) Correlation relationship between negative regulation DC score of all cell types and DCs' co-stimulatory score at sample level was showed by lollipop plot. r means correlation coefficient calculated by Pearson's correlation test, * $p < 0.05$.

D

E

Supplementary Figure 2E. (E) A dot plot was used to illustrate the upregulated receptor-ligand pairs in MTC compared to PTC, with the fold change on the right indicating the ratio of differences.

Figure 7B. Scatter plot showing the correlation relationship between CGRP expression and the Immune score, the cytotoxic score of CD8⁺T cell, the dysfunctional score of CD8⁺T cell and the co-stimulatory score of DCs in tumor bulk-RNA data. *r* indicates the correlation coefficient calculated by Spearman correlation test.

To validate the significant role of CGRP in the MTC, we initially used immunohistochemistry (IHC) to confirm the high expression of CGRP in MTC (Figure 2D). We observed that elevated CGRP expression was associated with poorer prognosis in MTC patients (Figure 2E). Furthermore, we employed multiplex immunohistochemistry (mIHC) to confirm the co-localization of CGRP secreted by MTC tumor cells with CGRP receptors on DC (Figure 2J).

Figure 2D-E. (D) In the tumor region of PTC and MTC, IHC staining for calcitonin and CGRP was shown. (E) Kaplan-Meier curve showing the disease-free survival (DFS) of MTC patients grouped

by the intensity of CGRP expression in the tumor region. MTC patients with high CGRP expression were characterized by yellow color while patients with low expression were characterized by blue color. *p*-value was calculated by log-rank test.

Figure 2J. mIHC showing the expression of CGRP, the percentage of CALCRL- positive CD11c+ cells in PTC and MTC tumor regions.

In vitro cell experiments were conducted to validate the inhibitory effect of CGRP on DCs. Firstly, we performed long-term experiments with CGRP treatment on CD8⁺ T cells, and no change was observed in the proliferation and cytotoxic functions of CD8⁺ T cells in CGRP exposure experiments (Supplementary Figure 7A-B), suggesting that CGRP did not directly affect the cytotoxic function of CD8⁺ T cells in MTC. Results from the CGRP-treated DC experiments revealed elevated levels of cAMP (Supplementary Figure 2G), increased expression of KLF2 (Figure 5C), and significantly weakened co-stimulatory function of DCs (Figure 5D and Supplementary Figure 5F), consistent with the characteristics of DCs observed in single-cell data of MTC. Furthermore, blocking CGRP signaling with CGRP receptor inhibitors or cAMP inhibitors restored KLF2 levels and functional molecule expression in DCs (Figure 5C-D and Supplementary Figure 5F).

Thus, our research results from genomic analysis and protein-level experiments demonstrated the primary and inhibitory role of CGRP in the MTC immune microenvironment and validated its inhibitory effect on DCs.

Supplementary Figure 7A-B. (A) Percentage of proliferating CD8⁺ T cells was shown under the treatment of CGRP. (B) MFI of IFN- γ and GZMB expression on mature CD8⁺ T cells. *p* value was calculated by one-way ANNOVAR compared to control group.

Supplementary Figure 2G. ELISA assay showed the concentration of cAMP (nM) after CGRP treatment, ** *p* < 0.01.

Figure 5C. Real-time PCR analysis of *KLF2* expression at day 6 during DC in different groups, ** $p < 0.01$, **** $p < 0.0001$.

Figure 5D. Representative flow cytometry histogram and increasing degrees of mean fluorescence intensity (MFI) of co-stimulatory markers CD80, CD40, CD86 and HLA-DR expressed on DCs. p values between groups were calculated by ANNOVAR, * $p < 0.05$, ** $p < 0.01$, *** $p < 0.001$, **** $p < 0.0001$.

Supplementary Figure 5F. Representative flow cytometry histogram and mean fluorescence intensity (MFI) of CD83 expression on mature DCs. *p* value was calculated by one-way ANNOVAR compared to control group, ****p*<0.001, *****p*<0.0001.

- What is the evidence of tumor and normal cell classifications? InferCNV results are blurry and color keys are not one-to-one mapped. The reference seemed to have tumor cells. Are MTC expected to have strong aneuploidy?

Our response: Thank you for your valuable question. We apologized for the unclear display of the InferCNV results and provided clear figures of CNV result in Supplementary Figure 1E-F. We reviewed the results and validated the differences in CNV signal correlation between reference cells and tumor cells using the method reported in previous study, in which the tumor cells often appeared in another peak of CNV correlation signal compared to reference cells (Puram SV., 2021, Cell). Additionally, we found that, as you mentioned, in the graph where parafollicular cells from adjacent tissues were used as a reference, there seemed to be a small portion of cells with CNV abnormalities from the cancer adjacent area, suggesting that these cells might have abnormal ploidy. This situation was reported by previous researches, which reported that both familial and sporadic MTC have observed certain parafollicular cell proliferation in normal thyroid tissue surrounding the tumor. This situation was named C-cell hyperplasia or micro-parafollicular lesions, occurring in the background of MTC in normal thyroid tissue, some of which may be precursors of MTC and exhibit cytological atypia (Perry A., 1996, Cancer; Kaserer K., 1998, Am J Surg Pathol). Therefore, considering the possibility of parafollicular proliferation and cytological changes in the reference cells, we considered them may be unsuitable as reference cells for CNV analysis. Hence, in our results, only T cells were retained as reference cell populations for CNV analysis of MTC and PTC tumor cells.

Supplementary Figure 1E-F. (E) The copy number alterations of malignant MTC cells inferred by inferCNV. T cells were selected as normal reference, and the copy number gain was shown in red color and the copy number loss were shown in blue color, similarly hereinafter. Density plot showed the correlation distribution of CNV signal among reference cells and MTC tumor cells. (F) The copy number alterations of PTC tumor cells inferred by inferCNV. T cells were selected as reference. Density plot showed the correlation distribution of CNV signal among reference cells and PTC tumor cells.

We have provided additional whole-exome sequencing data from paraffin-embedded sections for the patients undergoing single-cell sequencing and conducted analysis of ploidy. The results indicated that both MTC and PTC exhibit non-diploidy, but no significant differences in ploidy were observed between the two groups.

Box plot presented ploidy of MTC and PTC tumor samples from WES data and no significant difference was observed.

- Lines 135-154, what were statistics? Were the immune “cold” observations due to sampling bias? What are the relative immune cell proportions compared to other well-known immune “cold” tumor types?

Our response: Thank you for your suggestion. In Line 135-154, we presented the proportion of immune cells in the single-cell tumor data we collected. We performed CD45 staining in the tumor regions of 120 MTC patients to validate the tumor infiltration status. As shown in Figure 1G, most MTC tumors were characterized by limited immune infiltration, confirming that these results were not due to sampling bias. As you suggested, we have reanalyzed the data by incorporating publicly available single-cell data from three known "cold" immune tumors, including pancreatic cancer (Caronni N.,2023, Nature), glioblastoma (Krishna S.,2023, Nature), and prostate cancer (Hirz, T., 2023, Nature Commun). The results revealed that, compared to these three "cold" immune tumors, the proportion of immune cells in MTC remains relatively lower. Additionally, we have included other publicly available single-cell data from PTC (Wang T., 2023, Front Immunol; Lu L.,2023, J Clin Invest) and ATC (Gao R., 2021, Nat Biotechnol; Lu L.,2023, J Clin Invest) compared the immune infiltration status. It is evident that immune infiltration in MTC thyroid cancer remains at the lowest level compared to PTC and ATC.

Figure 1D. Cell type proportions of immune cells and non-immune cells. Each stacked bar represents one cancer type. Papillary thyroid cancer, PTC; Medullary thyroid cancer, MTC; Anaplastic thyroid cancer, ATC; Breast cancer, BC; Gastric cancer, GC; Hepatocellular carcinoma, HCC; Intrahepatic cholangiocarcinoma, ICC; Prostate cancer, PRAD; Pancreatic cancer, PDAC; Glioblastoma, GBM.

- The authors analyzed 8 MTC and 7 PTC tumors, however, there are multiple thyroid cancer datasets published, including scRNAseq of PTC and ATC, and bulk RNAseq of TCGA studies. What are the comparisons between the newly generated data to previously published larger cohorts?

Our response: Thank you for your suggestion. At single-cell data level, we incorporated multiple publicly available datasets of PTC (Wang T., 2023, Front Immunol; Lu L.,2023, J Clin Invest) and ATC (Gao R., 2021, Nat Biotechnol; Lu L.,2023, J Clin Invest), and transcriptomic data from the TCGA database to validate the immune microenvironment characteristics of MTC in our study. Firstly, as mentioned in our previous response, we included public thyroid cancer datasets for immune infiltration analysis. The results demonstrated that compared to PTC and ATC, MTC exhibits the lowest proportion of immune infiltration among these tumors (Figure 1D). Similarly, the results of immune infiltration scores at the transcriptomic level also indicated significantly lower immune infiltration scores in MTC compared to our PTC dataset and the TCGA PTC dataset (Figure 1F).

Figure 1F. Bar graph showing the predicted immune score of PTC (from our cohort and TCGA public dataset) and MTC in the bulk-RNA data, $**p < 0.01$.

Secondly, we compared the co-stimulatory scores of DC cells and the expression levels of *KLF2* among various thyroid tumors. The results indicated that among MTC, PTC, and ATC tumors, DCs in MTC exhibited the lowest co-stimulatory function scores (Supplementary Figure 4I), while displaying the highest levels of *KLF2* expression (Supplementary Figure 4J).

Supplementary Figure 4I-J. (I) Violin plot showing the genes signature score of co-stimulation in DCs derived from tumor of MTC, PTC and public dataset of PTC and ATC. Calculated by Wilcoxon test, $* p < 0.05$, $** p < 0.01$, $*** p < 0.001$. (J) Violin plot showing the genes expression of *KLF2* in DCs derived from tumor of MTC, PTC and public dataset of PTC and ATC. Calculated by Wilcoxon test, $*** p < 0.001$.

As depicted in Supplementary Figure 2F, consistent with our data, MTC DCs exhibited the highest activation scores in the cAMP pathway among all tumors, indicating significant activation of the cAMP pathway in MTC's DCs. When calculating the correlation between activation levels of cAMP pathway in DCs with *KLF2* expression levels or DC co-stimulatory function, we found higher cAMP pathway activation scores positively correlated with *KLF2* expression levels in DCs (Supplementary Figure 5B) and correlated negatively with co-stimulatory function (Supplementary Figure 5C) in DCs.

Supplementary Figure 2F. Violin plot showing the genes signature score of cAMP related pathways activation in DCs derived from tumor of MTC, PTC and public dataset of PTC and ATC. Calculated by Wilcoxon test, * $p < 0.05$, ** $p < 0.01$, *** $p < 0.001$.

Supplementary Figure 5B-C. (B) Scatter plot showing the correlation relationship between *KLF2* expression and cAMP activation level of tumor DCs when integrated with public single-cell dataset.

(C) Scatter plot showing the correlation relationship between DC co-stimulatory score and cAMP activation level of tumor DCs when integrated with public single-cell dataset.

Lastly, we also compared the functional status of CD8⁺ T cells in MTC, PTC, and ATC. As shown in Supplementary Figure 6I-J, MTC group exhibited the lowest scores in both cytotoxicity and dysfunction. Transcriptomic data similarly indicated lower cytotoxic and dysfunctional functions in MTC compared to the TCGA cohort of PTC (Supplementary Figure 6L-M).

Supplementary Figure 6I-J. Violin plot showing the genes signature score of the cytotoxicity and dysfunction in tumor CD8⁺ T derived from MTC, PTC and public dataset of PTC and ATC. Calculated by Wilcoxon test, *** $p < 0.001$.

Supplementary Figure 6L-M. The expression scores of the cytotoxic and dysfunctional gene signatures were calculated in the bulk-RNA data and presented in box plots, * $p < 0.05$, ** $p < 0.01$.

Reviewer #3: Major Comments:

- A key question is whether there is a direct interaction between CGRP and tumor-infiltrating T cells. It would benefit from cell experiments with CGRP and T cells.

Our response: Thank you for your suggestion. To investigate whether CGRP has a direct impact on tumor-infiltrating T cells, we conducted additional *in vitro* experiments involving CGRP and CD8⁺T cells. After isolating CD8⁺T cells from PBMCs, we cultured them for 4 days in the presence of CGRP alone or CGRP combined with a CGRP receptor inhibitor. Subsequently, we performed CFSE proliferation assays and flow cytometry analysis of CD8⁺T cell cytotoxicity marker IFN- γ and GZMB. The results showed that long-term treatment with CGRP did not affect CD8⁺T cell proliferation or the expression of functional molecules (Supplementary Figure 7A-B).

Supplementary Figure 7A-B. (A) Percentage of proliferating CD8⁺ T cells was shown under the treatment of CGRP. (B) MFI of IFN- γ and GZMB expression on mature CD8⁺ T cells. *p* value was calculated by one-way ANNOVAR compared to control group.

- How to determine the concentration of Rimegepant against the CGRP receptor?

Our response: Thank you for your question. For the selection of Rimegepant concentration, we referred from relevant study, which indicated that the safe dose of Rimegepant was below 10uM concentration, and 100nM Rimegepant can effectively restore CGRP-influenced function (Sun K., 2021, Oxid Med Cell Longey). Subsequently, we conducted preliminary experiments using

concentrations of 100nM, 200nM, and 400nM. Based on the lowest concentration that showed optimal restoration results, we chose a concentration of 200nM Rimegepant. This part has been added to the methodology section. (Page 43, Line 903-907)

- In Fig 1G, the samples used for IHC confirmation were not specified in the methods section. It would be better to provide the patients' characteristics of these samples.

Our response: Thank you for your suggestion. We have provided the clinical basic information of MTC patients for the IHC experiments in the Supplementary Table 7.

- In Fig 2E, were the same samples used in Fig 1G? How to calculate the score of CGRP expression?

Our response: Yes, in Fig 2E, we used samples with follow-up information from Fig.1G. We have supplemented relevant explanations in the methodology section. "Image J software was used to analyzed expression intensity of CGRP. We randomly selected 5 region of tumor and corrected the light density of each region we chose. Identified the stained-positive region by using segmentation in HIS mode, we selected measurement-IOD/ area (cm²) to calculate average of density (AOD) value. Kept the parameters unchanged and repeated all the steps for 5 regions and finally calculated the average of density of each sample." (Page 40-41, Line 840-846).

- Fig 4I did not show similar trends between MTC expressing CGRP and PTC without CGRP expression. This should be clarified.

Our response: Thank you for your suggestion. We have revised accordingly.

REVIEWERS' COMMENTS

Reviewer #2 (Remarks to the Author):

When public data are used, the data source/publications need to be cited. This was missing in multiple places. Please make sure to add citations to all analytic methods as well.

The manuscript didn't indicate if the tissue collection and the reported genomic studies were under IRB approval and received patient consents.

Reviewer #3 (Remarks to the Author):

The authors have revised the manuscript as possible as they can do according to the comments. It may be acceptable for publication.

Response to Reviewers' comments

Neurotransmitter calcitonin gene-related peptide shaped an immunosuppressive microenvironment in medullary thyroid cancer

We sincerely appreciate that the reviewers have provided helpful advices for manuscript refinement. Please find our point-by-point responses to the reviewers' comments as followed:

Reviewer #2:

1. When public data are used, the data source/publications need to be cited. This was missing in multiple places. Please make sure to add citations to all analytic methods as well.

Our response: Thank you for your suggestions. We have checked and cited the data source/publication in result and analytic method parts.

2. The manuscript didn't indicate if the tissue collection and the reported genomic studies were under IRB approval and received patient consents.

Our response: Thank you for your suggestions. We have indicated in Methods part. "The study was approved by the Institutional Research Ethics Committee of The First Affiliated Hospital of Sun Yat-sen University ([2021]109). The informed consents were collected from all patients for their willingness to participate in our project and may potentially publish clinical information that may identify individuals."

Reviewer #3:

The authors have revised the manuscript as possible as they can do according to the comments. It may be acceptable for publication.

Our response: Thank you for your thorough review and positive feedback. We sincerely appreciate your recognition of our efforts to address the comments and improve the manuscript.